**communications** engineering

# Exploring five types of beam shaping using tiled-aperture coherent beam combining

Yunhui Xie ✉, James A. Grant-Jacob , Matthew Praeger , Michalis N. Zervas & Ben Mills

Coherent Beam Combination (CBC) presents a promising solution to circumvent the power-scaling limitations of High-Power Fiber Lasers (HPFLs) by spatially combining the outputs of multiple independently pumped fibres. This parallel pumping configuration allows each fibre to operate below the critical threshold that would otherwise lead to instability, whilst their combined output exceeds the maximum power achievable from a stably operating HPFL. In this work, we demonstrate that manipulating the relative phases between fibre outputs extends the capabilities of CBC to approximate the phase profiles of optical elements such as spherical lenses, axicon lenses, and spiral phase plates, enabling versatile beam focus shaping and steering. We further show that the combined beam focus, whether coherently combined or shaped into a bespoke profile (e.g. Bessel-like, orbital angular momentum), can be steered and rotated in three-dimensional space through phase-only control. These results are experimentally validated through spatial light modulation to simulate collimated fibre outputs with controllable relative phases. Our findings advance CBC systems beyond mere power scaling, offering pathways for highly versatile beam shaping and steering, with implications for next-generation multifunctional optical power delivery systems.

Optical fibres allow light to be propagated inside confined, elongated, and flexible waveguides, and have found broad applications in areas of substantial economic and scientific interest, such as communication[1], sensing[2] and optical power generation and delivery[3]. Over the past decade, whilst considerable progress has been made in scaling up the output power of high-power fibre lasers[4–6], numerous thermo-optic and non-linear effects[7–9] (e.g. transverse mode instability[10,11], stimulated Brillouin scattering, and stimulated Raman scattering[12]), which could severely limit further power scaling, have also been identified and studied. In the face of these limitations, although current predictions suggest that the maximum power capability of a single-mode high-power fibre laser (using extremely large mode-area, Yb3+-doped fibres, tandem-pumped at 1018 nm and lasing at 1030 nm) can potentially reach approximately 80 kW[13], current state-of-the-art manufacturable diode-pumped fibre lasers provide stably single-mode, almost-diffraction-limited outputs in the range of 5 kW[14,15]. To circumvent these limitations, alternative approaches[16] (e.g. spectral beam combining[17–19]) have been pioneered, aiming to further scale up the power output[20], with Coherent Beam Combination (CBC)[21–23] emerging as one of the leading techniques. CBC leverages multiple, parallel active fibres to spatially combine their amplified outputs via a beam combiner. Whilst intuitively, this parallel combination scheme mitigates the power-scaling constraints imposed on individual fibres, it also introduces fluctuating phase noise that deteriorates the temporal coherence between the fibre outputs, thereby worsening the beam quality of the combined output. A key challenge in devising a CBC system is the suppression of fluctuating phase noise to maintain mutual coherence between fibre outputs.

Beyond power scaling capabilities, CBC systems have also been extensively studied for their potential as phased arrays, enabling further control over the combined beam focus, such as agile beam steering and beam shaping[24]. Commercially, *Civan Lasers* has demonstrated advanced laser welding applications utilising dynamic beam steering and shaping capabilities via CBC systems[25,26]. Although the technical details of these capabilities have not been fully disclosed, it is stated that they are achieved by treating the CBC system as an optical phased array[27]. The existing literature addresses several innovative extensions of CBC functionalities. Chang et al.[28] reported on modulating the phase profile of a CBC system to approximate a spaced dual-lens configuration, thereby achieving axial steering of the combined beam focus. Jabczyński[29] demonstrated the generation of Bessel beams using CBC, whilst various other studies have explored the generation of beams carrying Orbital Angular Momentum (OAM) in CBC system[30–36].

In this work, we propose a theoretical framework that leverages phase-only control of relative phases between CBC channels to approximate the phase profiles of transmissive, phase-delaying optical elements, such as lenses, axicons, and spiral phase plates. This approach allows the functionalities of these optical elements to be approximated by a CBC system functioning as a phased array, without requiring additional hardware. Our

Optoelectronics Research Centre, University of Southampton, Southampton, UK. ✉e-mail: Yunhui.Xie@soton.ac.uk

investigation begins with the phase-only approximation of a lens in a contacting dual-lens configuration, through which we demonstrate simultaneous steering of the combined beam focus along both axial (i.e. z) and transverse (i.e. x- and y-transverse) directions. This is accomplished by adjusting the effective focal length and applying lens decentring aberrations to a 'virtual' lens created by treating the CBC system as a phased array. This SLM-based 'virtual' lens is used in combination with a conventional lens to approximate a contacting dual-lens configuration. Furthermore, we extend this concept to demonstrate that three-dimensional steering capabilities are not limited solely to coherently combined focal spots but are also applicable to various structured beam profiles, including Bessel-like beams and beams with defined OAM states. Finally, we present a video demonstration showcasing simultaneous phase-only control of the combined beam focus, illustrating capabilities including: (1) varying the spatial position along axial (i.e. z) and transverse (i.e. x- and y-transverse) directions, (2) dynamic beam shaping through rotation of the transverse intensity distribution, and (3) controllable OAM, characterised by adjusting the rotation rate of the helical phase front with propagation.

In a tiled-aperture CBC configuration, as depicted in Fig. 1(a1), a centrosymmetric, close-packed array of collimated beams is spatially combined using a convex lens, $f_1$. This configuration allows the collimated beams to constructively interfere predominantly to a single spot at the principal focal plane of $f_1$, provided there are no relative phase differences between them. Introducing a second lens, $f_2$ (which can be either convex or concave), before the convex lens $f_1$ imposes additional phase delays to the collimated beams, thereby causing the focal plane to shift axially, either forward or backward. When $f_1$ and $f_2$ are treated as ideal thin lenses in contact, they can be modelled as a single lens with an effective focal length given by: $f_1 \cdot f_2 / (f_1 + f_2)$, as depicted in Fig. 1(a2). This modification changes the focusing properties of the beam combiner and shifts the focal plane at which constructive interference occurs, enabling axial steering of the combined beam focus.

Since the lenses $f_1$ and $f_2$ are assumed to be ideal thin lenses in contact, the beams between the two lenses experience only the phase delays imposed by the second lens $f_2$ before immediately reaching the first lens $f_1$. Therefore, the light fields of the beams have insufficient space to evolve radially or axially between the two lenses. That is, due to the negligible distance, the collimated beams do not undergo appreciable diffraction or spatial evolution between the lenses. This implies that the collimated beams remain collimated, preserving their transverse field distributions upon reaching both lenses. Consequently, the phase delays introduced by the second lens $f_2$ can be effectively approximated by intentionally offsetting the relative phases of the collimated beams (i.e. discretising the continuous phase profile). These phase offsets modulate the focusing properties of the beam combiner, enabling phase-only axial beam steering, as depicted in Fig. 1(a3). If the second lens $f_2$ is a simple spherical lens imposing quadratic phase delays, these delays can be approximated by fitting the phase profiles of the beams to a parabolic function, as illustrated in Fig. 1(b1), (b2). Alternatively, if the second lens $f_2$ is a Fresnel lens, the phase offsets can be sampled from the phase profile of the Fresnel lens, by taking the phase values at the geometric centres of the beams as illustrated in Fig. 1(c1), (c2). The physical meanings of both approaches are almost identical, as a Fresnel lens approximates a spherical lens by segmenting the continuous parabolic phase profile into discrete steps (i.e. wrapping the phase), and both approaches aim to impose a quadratic phase delay across the aperture, either continuously as shown in Fig. 1(b1) (i.e. spherical lens) or discretely as shown in Fig. 1(c1) (i.e. Fresnel lens). However, for visual clarity, the Fresnel lens approach is used throughout this work to present the phase profiles of the beams.

A Spatial Light Modulator (SLM) was employed to modulate a continuous-wave laser from an intensity-stabilised Helium-Neon source, simulating the collimated outputs of a hexagonally close-packed array of 61 fibres (approximated top-hat shaped with beam radii of 0.41 mm, $M^2 \approx 1.626$ for the coherently combined focus, approximately 76.3% fill factor). Due to this relatively low fill factor, side lobes may appear more prominently[37]. This setup enabled simultaneous control of both the intensity and the phase for each individual simulated fibre output. The simulated fibre outputs, after the SLM modulation, were first expanded and then spatially combined via a lens with a 50 cm focal length, corresponding to the first lens, $f_1$, as described earlier. To observe the spatial intensity distribution after the lens $f_1$, a camera was mounted on a motorised translation stage with a 45 cm travel range, allowing it to be moved along the beam propagation axis (i.e. axial direction). The camera directly observes the intensity distribution in the transverse plane (i.e. x–y transverse plane), whilst the intensity distribution along the axial direction is reconstructed by stitching together multiple transverse intensity distributions captured at discrete axial positions. All figures and discussions presented in this manuscript are based solely on experimental results, and all relevant raw data and code have been open-sourced (see Data availability statement and Code availability statement). Details of the experimental setup, including the simultaneous control of the phase and intensity of each simulated fibre output, and the phase offset calculations for approximating the second lens $f_2$ with varying focal lengths for axial combined beam focus steering, are provided in the Methods section. Although the SLM approach, in theory, should be able to simulate the combined output of an ideal tiled-aperture CBC system, real-world, fibre-based, high-power CBC systems rarely achieve such idealised outputs due to a range of practical complications. For instance, misalignments arising from tip–tilt and installation errors introduce wavefront aberrations, which must be accounted for during phase profile approximation. Moreover, high-power operation typically requires more complex system configurations, often involving additional amplification stages and active cooling components. These additions pose substantial challenges for managing thermal and environmental variations, which could compromise the performance of both phase control hardware and software. Lastly, the SLM-based approach implicitly assumes that beamlets remain perfectly collimated (i.e. diffraction-induced beam divergence and associated sidelobe growth can be neglected), whereas in a fibre-based, high-power, tiled-aperture CBC system each fibre output will undergo finite diffraction, leading to extra losses and increased sidelobe intensities. Together, these complications highlight the gap between the idealised SLM-based demonstration and a practical fibre-based, high-power CBC system, underscoring the need for further development to translate the proof-of-principle concept into a fibre-based experimental demonstration, which we aim to pursue in future work.

## Results
### Steering the combined focus in axial direction by approximating a lens

To steer the combined beam focus axially by $\epsilon$ from the principal focal plane of $f_1$, the focal length of the second lens, $f_2$, is given by: $f_2 = f_1 \cdot (f_1 - \epsilon) / [f_1 - (f_1 - \epsilon)] = f_1 \cdot (f_1/\epsilon - 1)$. Here, a positive scalar $\epsilon$ is defined as steering the combined beam focus towards the lens $f_1$ by $\epsilon$ (i.e. shortening the effective focal length by $\epsilon$ using a convex $f_2$), whilst a negative scalar $\epsilon$ is defined as steering the focus away from the lens $f_1$ by $\epsilon$ (i.e. lengthening the effective focal length by $\epsilon$ using a concave $f_2$). The additional phase delays imposed by the second lens, $f_2$, can thus be calculated and approximated by offsetting the relative phases of beams, thereby enabling phase-only axial beam focus steering. Fig. 2(a1) presents the intensity distribution of the combined beam focus along the axial direction from 10 cm before to 10 cm after the principal focal plane of $f_1$ (which accounts for 40% of the first lens' focal length). The combined beam focus is dynamically steered to each axial position by approximating $f_2$ with the corresponding focal length. The transverse intensity distributions of the combined beam focus at 5 cm before the principal focus of $f_1$, at the principal focus of $f_1$, and 5 cm after the principal focus of $f_1$ are provided in Fig. 2(a2), (a3) and (a4), respectively. For comparison, Fig. 2(b) provides the intensity distribution of the beams along the axial direction without phase offsets (i.e. in-phase), with corresponding transverse intensity distributions 5 cm before the principal focus of $f_1$, at the principal focus of $f_1$, and 5 cm after the principal focus of $f_1$ shown in Fig. 2(b2), (b3) and (b4), respectively. The comparison clearly demonstrates that the combined beam focus can be

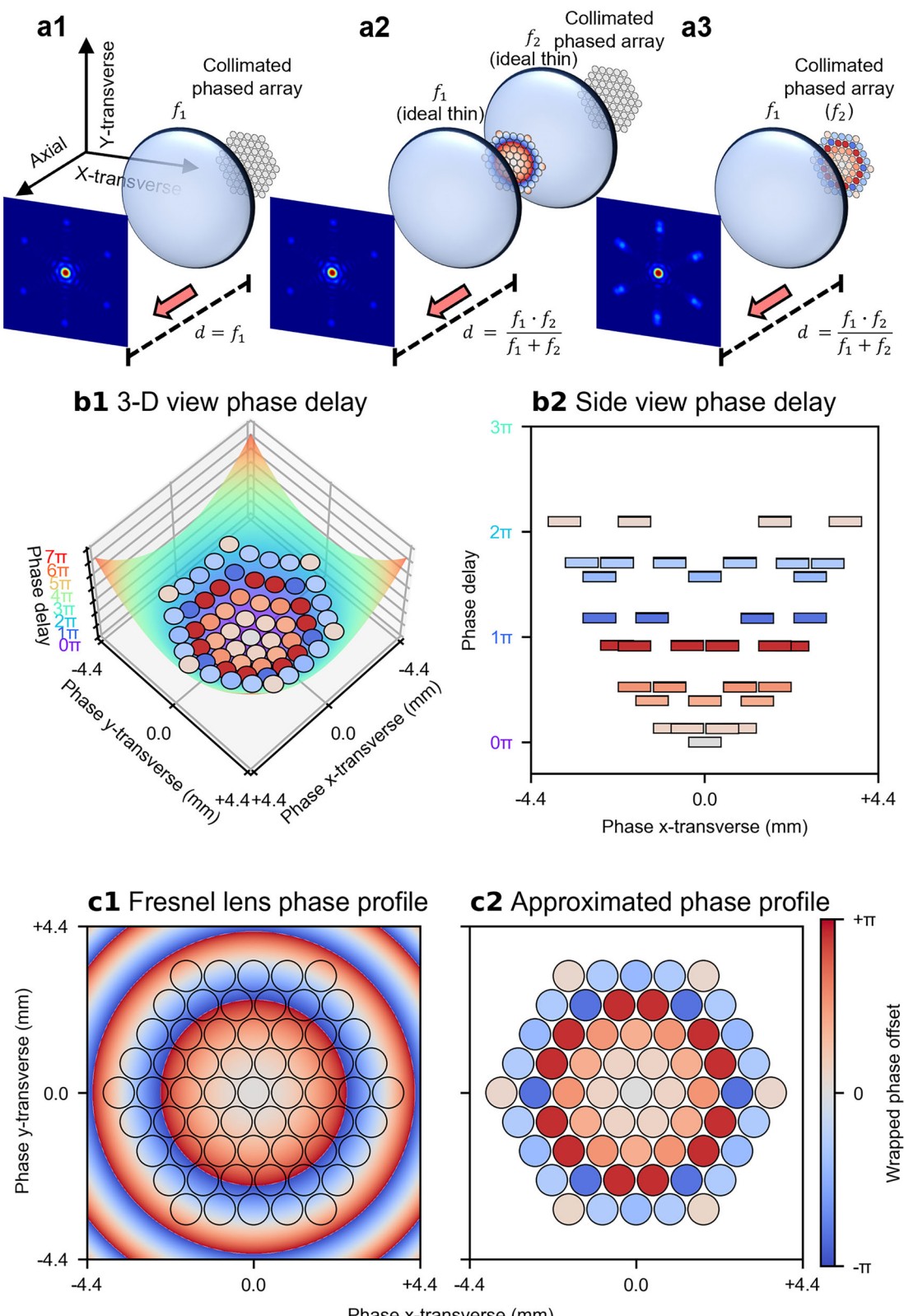

**Fig. 1 | Concept of the phase-only CBC combined beam focus steering.**
**a1** Coherent combination of 61 collimated beams in a tiled-aperture CBC system, where a convex lens $f_1$ focuses (combines) the collimated beams at its principal focal plane. **a2** Inserting a second lens $f_2$ (either convex or concave) before $f_1$ shifts the focal plane axially, moving the coherently combined beam focus of the collimated beams either backward or forward depending on the lens specs of $f_2$. **a3** Simulating the phase delays imposed by inserting an ideal thin lens $f_2$ through offsetting the

relative phases of the beams results in phase-only axial beam focus steering. **b1** The phase delays imposed by a lens can be described by a parabolic function; a corresponding side view is shown in (**b2**). **c1** Similarly, inserting a Fresnel lens shifts the focal plane and the combined beam axially (**c2**), The phases of the beams are sampled from the phase offsets imposed by a Fresnel lens via taking the phase values at the geometric centres of the beam locations.

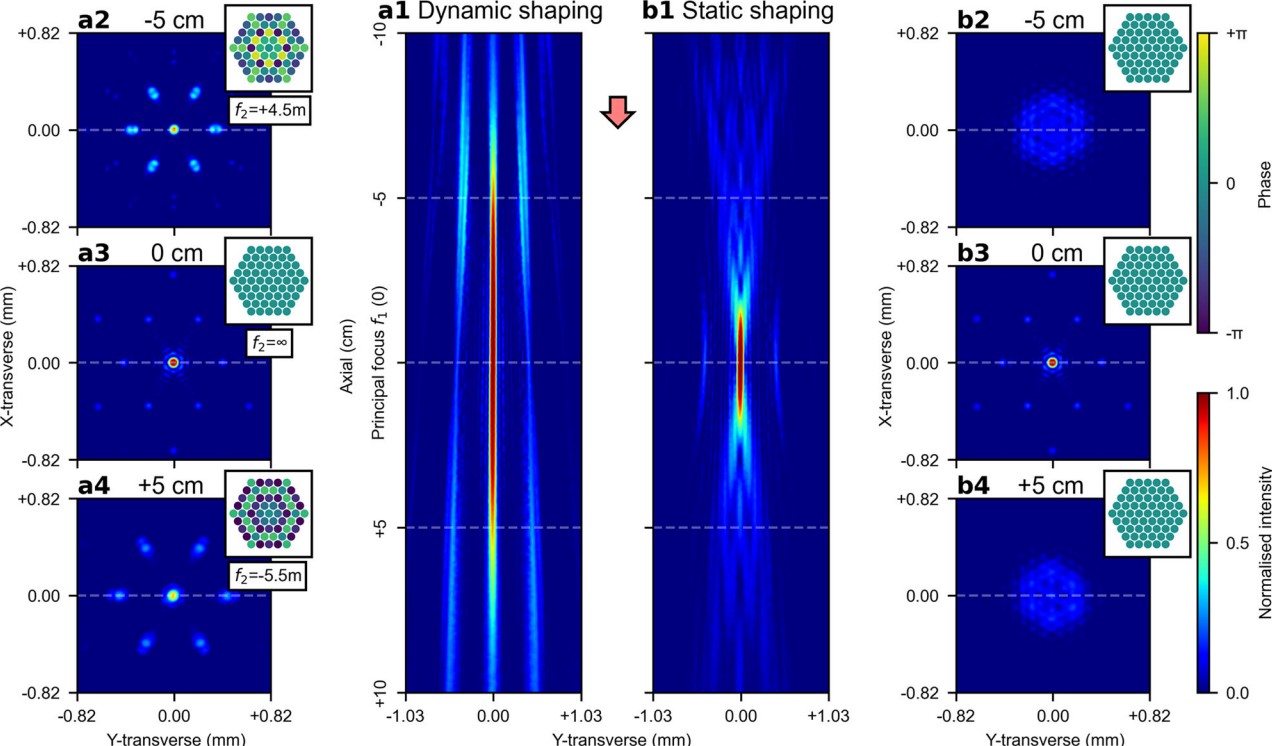

**Fig. 2 | Phase-only beam focus steering in the axial direction. a1** Axial beam steering by dynamically offsetting the phases of the beams, effectively approximating the phase delays imposed by the second lens, $f_2$, with varying focal lengths. Here, at each axial position, the relative phases of beams are offset to approximate $f_2$ with the corresponding focal length, thereby steering the combined beam focus to the target axial position. The transverse intensity distributions (i.e. x-y transverse plane) of the combined beam focus are shown at three axial positions: 5 cm before the principal focus of $f_1$ (labelled as −5 cm), at the principal focus of $f_1$, and 5 cm after the principal focus of $f_1$ (labelled as +5 cm), as presented in (**a2**), (**a3**), and (**a4**), respectively. To shift the combined beam focus by ±5 cm, the relative phases of beams approximate the additional phase delays imposed by the second lens $f_2$ with focal lengths of 4.5 m for −5 cm and −5.5 m for +5 cm, respectively. **b1** Comparatively, the intensity distribution along the axial direction is shown without phase offsets between the beams. The corresponding transverse intensity distributions of the combined beam 5 cm before the principal focus of $f_1$, at the principal focus of $f_1$, and 5 cm after the principal focus of $f_1$ are presented in (**b2**), (**b3**) and (**b4**), respectively.

steered axially by offsetting the relative phases of the beams. However, it can also be observed that as the combined beam focus is steered further from the principal focal plane of $f_1$, the transverse intensity distribution deviates more appreciably from the transverse intensity distribution at the principal focal plane of $f_1$. This can be largely attributed to the fact that steering the combined beam focus farther away requires progressively smaller focal lengths for $f_2$ (i.e. $f_2 \propto 1/\epsilon$), thereby leading to increasingly larger phase delays imposed by $f_2$ in the limited aperture. These larger phase delays render offsetting the phases less effective at approximating $f_2$. A more detailed analysis of the approximation ability and its impact on the power-in-bucket metric is provided in the Supplementary Information Section A and Fig. S1. It is also noteworthy that, mathematically, $f_2$ becomes undefined when $\epsilon = 0$, representing a singularity where the focal length of $f_2$ is infinite. Physically, in order to steer the combined beam focus to the focal point of $f_1$ (where $\epsilon = 0$), the second lens $f_2$ imposes uniform phase delays to all beams (i.e. focal plane at infinite distance), implying that no phase offsets are needed for the beams to approximate $f_2$ at $\epsilon = 0$.

### Steering the combined focus in transverse direction by introducing lens decentring

Similar to the approach of approximating a lens for axial steering, the relative phases of the beams can also be offset to approximate a wedge prism, thereby enabling steering of the combined beam focus along the transverse directions (i.e. x-transverse and y-transverse). Alternatively, phase offsets can be also sampled from the phase delays imposed by a transmission grating, which functions equivalently to a wedge prism (i.e. imposing planar, linear phase delays). This phase-only steering along the transverse directions

can be combined with the previously discussed phase-only axial steering by imposing additional phase delays before the second lens, $f_2$, enabling complete phase-only combined beam focus steering in the three-dimensional space (i.e. x-, y-transverse, and axial). A different, and perhaps more intriguing, approach for achieving phase-only, three-dimensional steering, without the need for approximating a third optical component, is to approximate a decentred $f_2$ using phase offsets. Decentring the second lens, $f_2$, induces monochromatic aberration that tilts the combined beam focus off the optical axis, thereby resulting in a displacement along the transverse directions relative to the principal focus of $f_1$. Fig. 3(a1), (b1) present the intensity distributions of the combined beam focus along the axial direction, from 10 cm before to 10 cm after the principal focal plane of $f_1$. In Fig. 3(b1), only axial steering is applied, whereas Fig. 3(a1) presents the simultaneous application of both axial and transverse steering. In the latter case, the second lens, $f_2$, is decentred vertically by 1.46 mm at each axial position. The transverse intensity distributions 3 cm before, at, and 3 cm after the principal focal plane of $f_1$ are shown in Fig. 3(a2), (a3), (a4) for simultaneous axial and transverse steering, with corresponding phase profiles of the decentred $f_2$ provided in Fig. 3(a5), (a6), and (a7), respectively. For comparison, Fig. 3(b2), (b3), and (b4) present the transverse intensity distributions for axial-only steering at the same axial positions.

Unlike the wedge prism approach, where the tilted combined beam focus immediately diverges away from the optical axis after leaving the first lens, $f_1$, in the decentred lens approach, the tilted beam first converges towards the optical axis before diverging beyond the principal focal point of $f_1$. As previous discussed, when $\epsilon = 0$ (i.e. the combined beam focus

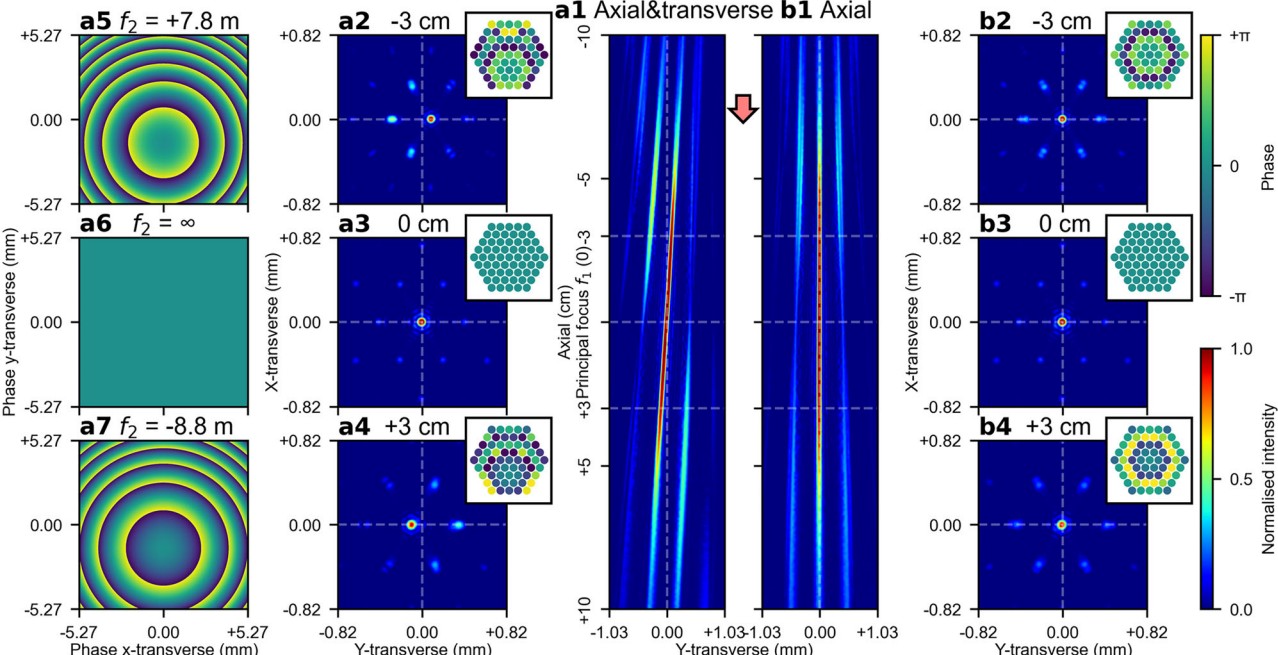

**Fig. 3 | Phase-only beam focus steering in both the transverse and the axial directions. a1** Simultaneous steering of the combined beam focus in both the axial and the transverse directions. The axial steering is achieved by approximating the second lens, $f_2$, with varying focal lengths through phase offsetting, whilst the transverse steering is achieved by vertically decentring $f_2$ by 1.46 mm. The transverse intensity distributions of the combined beam are shown at three axial positions: 3 cm before the principal focus of $f_1$, at the principal focus of $f_1$, and 3 cm after the principal focus of $f_1$, as presented in (**a2**), (**a3**), and (**a4**), respectively. The corresponding phase profiles for $f_2$ from which the phase offsets are sampled, are presented in (**a5**), (**a6**), and (**a7**). **b1** As a comparison, axial steering of the combined beam focus without any lens decentring, with the transverse intensity distributions shown at 3 cm before the principal focus of $f_1$, at the principal focus of $f_1$, and 3 cm after the principal focus of $f_1$ in (**b2**), (**b3**), and (**b4**), respectively.

coincides with the principal focal point of $f_1$), the reciprocal relationship between $\epsilon$ and $f_2$ leads to an infinite focal length for $f_2$, resulting in uniform phase delays imposed to all beams. These uniformly imposed phase delays prohibit lens decentring from steering the combined beam focus along the transverse directions at $\epsilon = 0$, ensuring that the tilted beam always passes through the principal focal point of $f_1$ irrespective of the decentring applied to $f_2$. The decentred lens and wedge prism approaches can also be combined to provide enhanced control over the combined beam focus steering along the transverse directions. An example provided in the Supplementary Information Section B and Fig. S2 demonstrates the simultaneous application of these two approaches in opposing tilt directions, which effectively cancels the beam tilting whilst maintaining displacement of the combined beam focus relative to the principal focal point of $f_1$ along the transverse directions.

### Steering shaped beam focus in both axial and transverse directions

A lens, acting as a beam combiner, performs a spatial Fourier transform of the beams in the far field. When there are no relative phase differences between the beams, the resulting spatial Fourier transform is dominated by a few spatial frequencies, leading to a concentrated intensity distribution at the Fourier plane after the lens beam combiner. The intensity distribution at the Fourier plane is determined by both the spatial frequencies introduced by the phase offsets between the beams and the effective focal length of the lens pair beam combiner (i.e. $f_1$ and $f_2$). Specifically, the shape of the intensity distribution is determined by the spatial frequencies (i.e. the phase offsets between beams), whilst the size and scaling of the intensity distribution are determined by the effective focal length. This implies that the previously described three-dimensional, phase-only combined beam focus steering can be applied to beams with any arbitrary phase offsets, as the shape of the intensity distribution at the Fourier plane (i.e. effective focal plane) is determined by the phase offsets of the beams rather than the focal length of the lens pair beam combiner.

Fig. 4(a1), (b1) present the intensity distributions of a shaped beam focus (as shown in Fig. 4(a3), (b3)) along the axial direction, from 10 cm before to 10 cm after the principal focal plane of $f_1$. In both figures, the shaped beam foci are steering along the axial direction. In the intensity distribution shown in Fig. 4(a1), additional lens decentring of 1.46 mm is applied to the second lens, $f_2$, at every axial position, which enables simultaneous steering along both the transverse and axial directions, whilst no lens decentring is applied to the intensity distribution shown in Fig. 4(b1). The transverse intensity distributions at three axial positions, namely 3 cm before, at, and 3 cm after the principal focus of $f_1$, are shown in Fig. 4(a2), (a3) and (a4), respectively, for the former case, and in Fig. 4(b2), (b3), and (b4), respectively, for the latter case. At the principal focus plane of $f_1$ (i.e. Fig. 4(a3) and (b3)), the transverse intensity distributions are identical to the shaped beam focus being steered due to the singularity at $\epsilon = 0$. This is because, when $\epsilon = 0$ (where $f_2 = \infty$), the second lens, $f_2$, imposes uniform phase delays (hence no phase differences) to all beams. A further example, demonstrating a shaped beam focus with a different set of phase offsets, is provided in the Supplementary Information Section C and Fig. S3.

### Steerable Bessel beam and beam carrying orbital angular momentum

Lastly, two bespoke beam shapes of considerable research interest, namely the Bessel beam and the Laguerre-Gaussian (LG) beam carrying OAM, are demonstrated. Traditionally, generating these beams requires one or more optical components, such as SLMs, axicon lenses or phase plates, which necessitates careful consideration of damage thresholds, particularly for high-power laser applications. However, the CBC phase offsetting approach presented in this work enables the generation of these beams without requiring additional optical components, thereby lessening concerns related to component damage thresholds. This approach also allows for beam steering in both axial and transverse directions, offering greater flexibility and reducing system complexity.

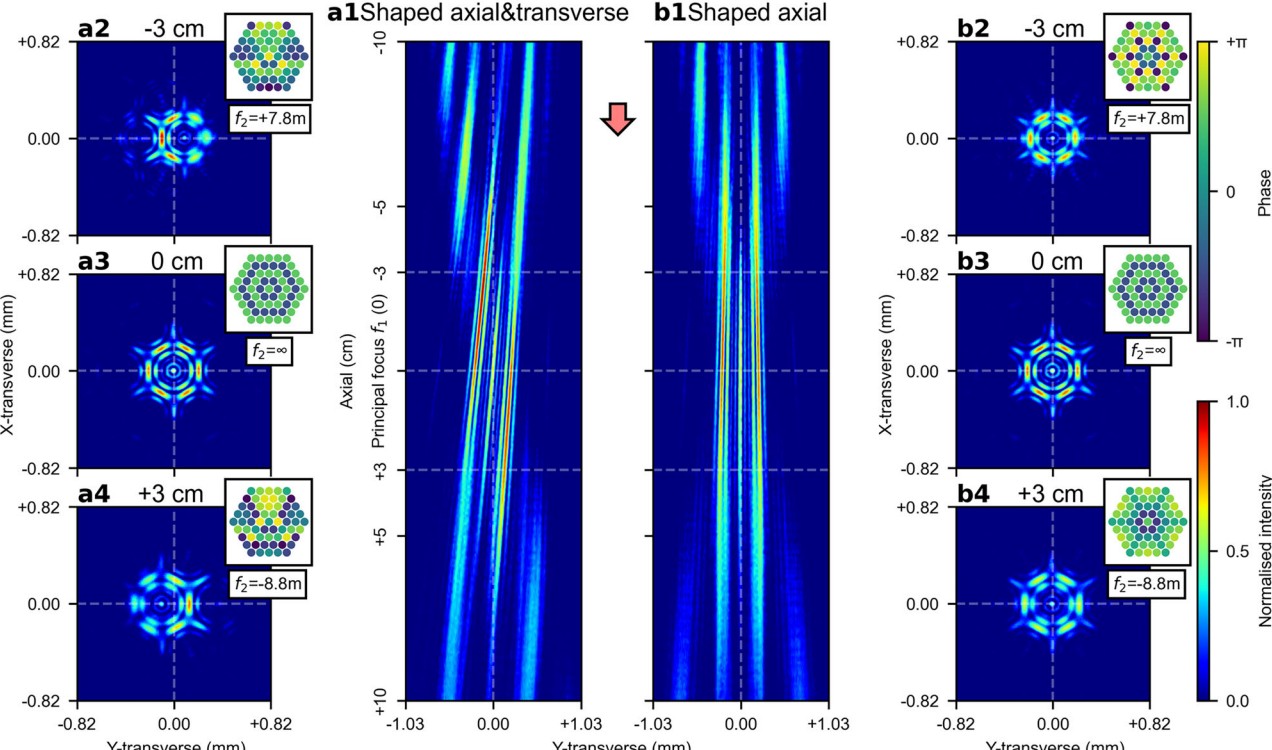

**Fig. 4 | Phase-only shaped beam focus steering in both the transverse and the axial directions. a1** Simultaneous steering of a shaped beam focus in both the axial and the transverse directions. The transverse intensity distributions at three axial positions, 3 cm before, at, and 3 cm after the principal focal plane of $f_1$, are shown in (**a2**), (**a3**), and (**a4**), respectively. **b1** Axial-only steering of the same shaped beam focus. The transverse intensity distributions at the same axial positions are shown in (**b2**), (**b3**), and (**b4**), respectively.

To generate a Bessel beam, an axicon lens is approximated by offsetting relative phases of beams, which imposes conical, linear phase delays (shown in Fig. 5(a1)), as opposed to the parabolic, quadratic phase delays imposed by a simple spherical lens. The phase profiles of the beams used to approximate the axicon lens via offsetting the relative phases of the beams are presented in Fig. 5(a2). The resulting Bessel-like intensity distribution along the axial direction is shown in Fig. 5(a4). Here, the axial position of the Bessel-like beam is defined as the axial position at which the peak intensity of the transverse intensity distribution is maximised, as depicted in Fig. 5(a3). To enable also the combined beam focus steering, two ideal thin optical components need to be modelled, namely the axicon lens and the second spherical lens, $f_2$, (either convex or concave). The phase offsets for the beams passing through these two optical components before reaching the first lens $f_1$ are calculated by considering the axicon lens and the second spherical lens $f_2$ as a single optical component, as shown in Fig. 5(b1), where the conical and the parabolic phase delays are superimposed. The phase offsets for beams are shown in Fig. 5(b2), with the corresponding axial and transverse intensity distributions presented in Fig. 5(b4), (b3), respectively. Comparing Fig. 5(a4), (b4) reveals an axial beam displacement of approximately 1.7 cm, whilst maintaining the characteristic intensity distribution of the Bessel-like beam.

This approach can also be extended to spiral phase plates, which impose phase delays along the azimuthal direction, thereby generating beams carrying OAM[38]. Fig. 5(c1), (d1) present the phase profiles for generating LG modes $LG_{02}$ and $LG_{12}$, respectively. These phase profiles are approximated via offsetting relative phases of beams, as shown in Fig. 5(c2), (d2), with corresponding transverse and axial intensity distributions shown in Fig. 5(c4), (d4), (c3) and (d3). Superimposing spiral phase plates on lenses with a focal length of 12 m (Fig. 5(e1), (f1) for $LG_{02}$ and $LG_{12}$, respectively) can be similarly approximated by offsetting relative phases of the beam, presented in Fig. 5(e2), (f2). The resulting transverse and axial intensity distributions are shown in Fig. 5(e3), (e4), (f3) and (f4). An axial steering of

approximately 2 cm is observed between the two LG beams with the same mode, achieved without noticeable distortion to the shaped beam focus. Moreover, in addition to steering in axial, x-transverse, and y-transverse directions, modulating the azimuthal phase delays enables rotational control of higher-order LG modes. This rotational capability, combined with full three-dimensional beam focus steering and LG mode control, constitutes a five-dimensional, phase-only control of the combined beam focus, as demonstrated in Supplementary Movie. Furthermore, the superimposition of a spiral phase plate on an axicon lens (i.e. diffractive spiral axicon) can enable the generation of a vortex beam, as detailed in the Supplementary Information Section D and Fig. S4. The Supplementary Information also includes fork-shaped interference patterns (see Supplementary Information Section E and Fig. S5), which determine topological charges of OAM beams, between combined beam focuses carrying different OAM modes and a reference beam[39]. Further discussion and additional results on the axial and transverse tuneability of beams carrying OAM modes are provided in Supplementary Information Section F.

## Conclusions
In this work, we have demonstrated a versatile method for modulating the combined beam focus in a simulated CBC setup. By introducing appropriate phase offsets between coherently combined beams, we effectively approximated transmissive, phase-delaying optical elements. This approach enabled three-dimensional steering of the combined beam focus, achieved by simulating the effects of a decentred spherical lens imposing off-optical-axis, parabolic, and quadratic phase delays. We discussed that the additional phase delays shift the Fourier plane of the beam combiner axially, thereby allowing a shaped combined beam focus to be steered. Furthermore, by approximating axicon lenses, which impose conical, linear phase delays in the radial direction, and spiral phase plates, which impose phase delays along the azimuthal direction, we demonstrate generations of Bessel-like beams and beams carrying OAM, respectively. Finally, we demonstrated a

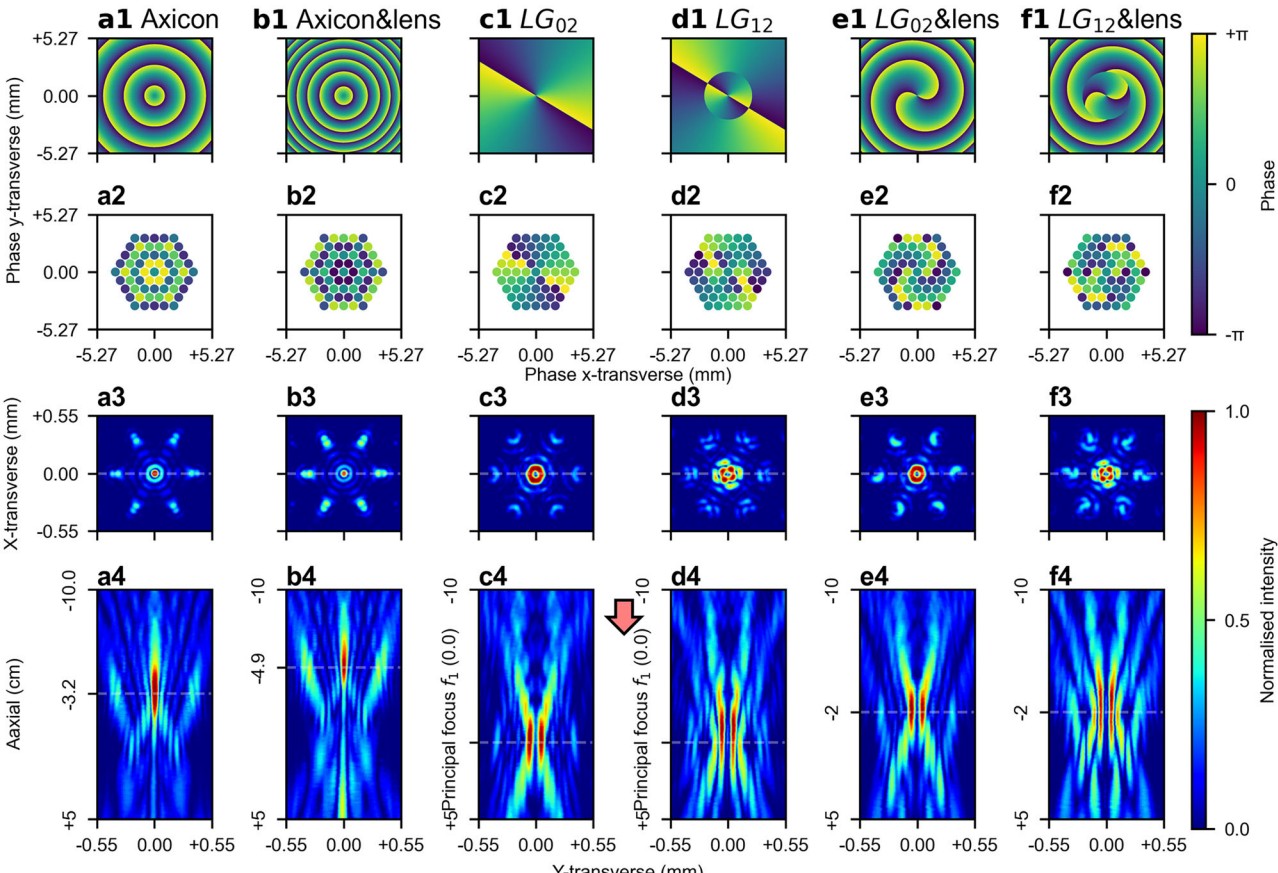

**Fig. 5 | Phase-only steerable Bessel beam and beam carrying OAM. a1** A wrapped conical, linear phase profile, which is physically identical to the phase delays imposed by an axicon lens, is discretised, and approximated by phase offsets of beams, as shown in (**a2**). These phase offsets generate a Bessel-like beam, presented in (**a4**), with the corresponding transverse intensity distribution with the maximum peak intensity, presented in (**a3**). **b1** The same conical, linear phase profile is super-imposed with a parabolic, quadratic phase profile from a simple spherical lens with a 12 m focal length. This combined phase profile is similarly approximated by phase offsets of beams, presented in (**b2**), resulting in a similar Bessel-like beam, presented in (**b4**). However, the additional phase offsets from the lens steer the beam axially, as indicated by the transverse intensity distribution with the maximum peak intensity shown in (**b3**). The two axial positions (i.e. (**a3**) and (**b3**)) with peak intensities are

separated by approximately 1.7 cm. **c1** and **d1** Phase profiles for the $LG_{02}$ and $LG_{12}$ spiral phase modes, respectively, are shown with their corresponding discretised phase offsets presented in (**c2**) and (**d2**). The transverse intensity distributions at the principal focus plane of $f_1$ (i.e. (**c3**) and (**d3**)) and axial intensity distributions (i.e. (**c4**) and (**d4**)) show the resulting beam profiles for each mode along the axial and transverse directions. (**e1**) and (**f1**), The phase profiles of the $LG_{02}$ and $LG_{12}$ modes are superimposed with a Fresnel lens phase profile (focal length 12 m), which modifies the focusing properties of the beam combiner. The corresponding fibre phase offset approximations are shown in (**e2**) and (**f2**), with the resulting transverse intensity distributions presented in (**e3**) and (**f3**), and axial intensity distributions presented in (**e4**) and (**f4**), where the shaped combined beam focus is moved 2 cm axially.

five-dimensional control over the combined beam focus, encompassing three-dimensional combined beam focus steering, rotational and bespoke shape control, all through phase-only modulation, which we presented in a video format. These results provide a complementary perspective, demonstrating how relative phase differences in a CBC system can be leveraged to approximate transmissive, phase-delaying optical elements. This perspective unlocks opportunities for comprehensive combined beam focus shaping and steering within CBC systems, without the need for additional hardware.

## Methods

A linearly polarised, Gaussian-shaped, intensity-stabilised, continuous-wave Helium-Neon laser (632.992 nm centre wavelength, 1.2 mW output power, *Thorlabs* HRS015B) was attenuated, expanded, and collimated before being directed onto a reflective Liquid-Crystal-on-Silicon Spatial Light Modulator (LCoS-SLM, *Thorlabs* EXULUS-HD1/M, referred to as SLM hereafter) via a non-polarising cube Beam Splitter (BS1, 50:50 split ratio) at normal incidence. The SLM applied blazed-grating-like periodic phase modulations, encoding both phase and intensity information into the first-order diffracted beam. The zeroth-order diffracted beam was reflected

to BS1 at normal incidence, whilst beams at higher diffraction orders, including the phase-and-intensity-encoded, first-order beam, were reflected at specific angles. These beams were directed onto a pair of mirrors via a second non-polarising cube beam splitter (BS2, 50:50 split ratio) to correct for angular deviations of the first-order beam before passing through a selective diffraction order suppression system[40]. This system, comprising two convex lenses, $F_1$ and $F_2$, and an iris, acted as a 4f system performing a cascade of two Fourier transforms. The iris, positioned at the Fourier plane, served as a spatial frequency filter, blocking all but the spatial frequencies associated with the first diffraction order. Specifically, the convex lens $F_1$ performed the initial spatial Fourier transform, mapping beams of different diffraction orders from the spatial domain to the spatial frequency domain. At the Fourier plane, the iris selectively transmitted the spatial frequencies corresponding to the first-order beam, whilst suppressing those from other diffraction orders (i.e. spatial frequency mask). The following lens $F_2$, then performed the inverse spatial Fourier transform, restoring the phase-and-intensity-encoded first-order beam from the spatial frequency domain back to the spatial domain. In addition to spatial filtering, this 4f system also magnified the spatially filtered beam by 1.5 times, due to the focal lengths of $F_1$ (50 cm) and $F_2$ (75 cm), thereby alleviating the size constraint imposed

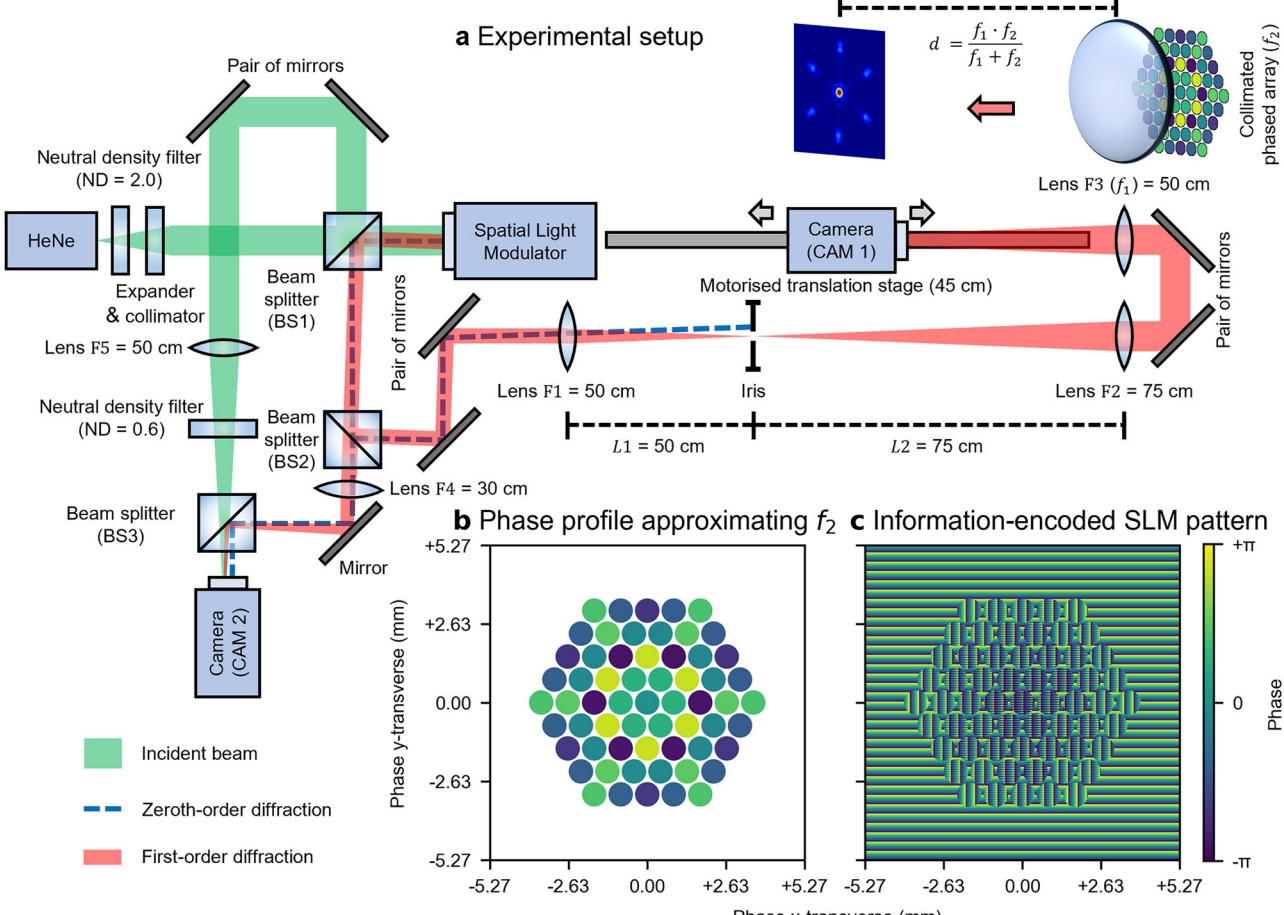

**Fig. 6 | Experimental setup, and simulating CBC fibre channels with controllable relative phases and intensities using a single SLM. a** Schematic of the experiment setup. **b** Phase profile of the collimated beams, which approximates the convex second lens $f_2$ with a focal length of 4.5 m. **c** Pattern displayed on the active area of the SLM. Note that the pattern shown in (**c**) already accounts for the magnification resulting from the selective diffraction order suppression system. Consequently, the 1920 × 1080-pixel active area on the SLM with the physical size of 12.48 mm ×

7.02 mm is magnified to 18.72 mm × 10.54 mm. The radii of simulated fibre outputs are approximately 0.41 mm (42 pixels). In an actual fibre CBC setup, the radii of collimated fibre outputs could be larger than the demonstrated values here, thereby limiting the minimum focal length of the approximated second lens $f_2$ as detailed in the Supplementary Information Section A. Thus, in this case, the focal length of the first lens $f_1$ must be adjusted accordingly from $f_1 \cdot (f_1/\epsilon - 1) - f_2 = 0$ to maintain the same distance $\epsilon$ for axial combined beam focus steering.

---

by the limited active area of the SLM. After the spatial filtering and magnification, the beam was redirected by a second pair of mirrors and focused by a final lens, $F_3$ (50 cm focal length), which served as the first lens $f_1$ discussed in the Main section. A camera (*Basler* a2A4504−18ucBAS, CAM 1), mounted on a motorised linear stage (*ZABER* X-LSQ450B-E01, 45 cm travel), was positioned facing the focused beam along the beam propagation direction (i.e. axial direction) and was moved to observe the intensity distribution of the focused beam at different axial positions. A schematic of the experiment setup is presented in Fig. 6(a).

The initial beam splitter (BS1) split the incident beam, directing one portion to the SLM for spatial light modulation, whilst the other portion served as a planar-wave reference beam for interference with the modulated first-order beam carrying OAM. This reference beam, after leaving BS1, was redirected by a pair of mirrors, passed through a convex lens $F_5$ (50 cm focal length) and a Neutral Density (ND, 0.6 optical density) filter, and then combined with the modulated OAM beam at a third non-polarising beam splitter (BS3, 50:50 split ratio). The second beam splitter (BS2), placed after the initial beam splitter (BS1), split the modulated first-order beam, directing one portion to the selective diffraction order suppression system and redirecting the other portion to a convex lens $F_4$ (30 cm focal length). This focused, modulated beam was then combined with the reference beam at BS3. The slight angular difference between the modulated beam and the

reference beam introduced planar, linear phase differences (i.e. off-axis interference), which, combined with azimuthal phase delays inherent in the OAM beam, resulted in fork-like interference patterns[39]. These patterns were captured by a stationary camera (*Basler* a2A4504-18ucBAS, CAM 2), positioned to observe the interference patterns at the principal focus of $F_4$. The fork-shaped interference patterns between OAM beams with different modes and the reference beam are shown in the Supplementary Information Section E and Fig. S5.

The phase transformation function of an ideal thin lens under the paraxial approximation is expressed as:

$$T_{lens}(x, y) = \exp\left[-i \cdot \frac{k}{2f} \cdot (x^2 + y^2)\right] \quad (1)$$

where $k = 2\pi/\lambda$ is the wave number, $f$ is the focal length of the second lens $f_2$, and $(x, y)$ are the transverse coordinates. To account for lens decentring, the transverse coordinates $(x, y)$ can be modified to $(x_0 - x_{off}, y_0 - y_{off})$, where $(x_0, y_0)$ are the transverse coordinates along the optical axis, and $x_{off}$ and $y_{off}$ represent the lens decentring along the x- and y-transverse directions, respectively. However, for simplicity, this expanded form will not be used in subsequent equations. The phase profiles $\phi_{out}(x, y)$ of the collimated beams after passing through an ideal thin lens, given an initial

phase profile $\phi_{in}(x, y)$ can be expressed as:

$$\phi_{out}(x, y) = Angle\left[T_{lens}(x, y) \cdot \exp\left(i \cdot \phi_{in}(x, y)\right)\right] \quad (2)$$

Here, the initial phase profile $\phi_{in}(x, y)$ can be represented as the superposition of the individual phase profile $\phi_{in}^i(x, y)$ from each beam, centred at $(x^i, y^i)$: $\phi_{in}(x, y) = \sum_n^i \phi_{in}^i(x, y)$. Assuming each beam has a circular profile with identical radius $r$ and a uniform phase value $\varphi^i$, the initial phase profile for an individual beam can be written as:

$$\phi_{in}^i(x, y) = \begin{cases} \varphi^i, & (x - x^i)^2 + (y - y^i)^2 \leq r^2 \\ 0, & elsewhere \end{cases} \quad (3)$$

Since the first lens, $f_1$, and the second lens, $f_2$, are assumed to be ideal thin lenses in contact, the light field immediately reaches the first lens upon leaving the second lens, preventing any transverse field evolution. Consequently, $\phi_{out}(x, y)$ can be similarly described as the superposition of the phase profiles $\phi_{out}^i(x, y)$ for all collimated beams: $\phi_{out}(x, y) = \sum_n^i \phi_{out}^i(x, y)$, where

$$\phi_{out}^i(x, y) = \begin{cases} \phi_{out}(x, y), & (x - x^i)^2 + (y - y^i)^2 \leq r^2 \\ 0, & elsewhere \end{cases} \quad (4)$$

However, unlike the initial phase profile $\phi_{in}^i(x, y)$, the phase profile after the lens $\phi_{out}^i(x, y)$ is no longer uniform, due to the quadratic phase term imposed by the lens. To approximate $\phi_{out}^i(x, y)$ using $\phi_{in}^i(x, y)$, the phase at the geometry centre of each collimated beam $\phi_{out}^i(x^i, y^i)$ is sampled and assigned as the initial phase value $\varphi^i$. Fig. 6(b) presents this approximation, showing the relative phase offsets of the collimated beams, with an initial phase $\varphi^i = 0$ for all fibres before the second lens $f_2$ (4.5 m focal length). To approximate the conical, linear phase delays imposed by an axicon lens, the phase transformation function is defined as: $T_{axicon}(x, y) = \exp[-i \cdot k_\alpha \cdot (x^2 + y^2)^{1/2}]$, where the $k_\alpha \in \mathbb{R}^+$ (i.e. positive real numbers) governs the slope of the linear phase delays, analogous to the axicon angle $\alpha$. Similarly, to approximate phase delays along the azimuthal direction from a spiral phase plate, the phase transformation function is given by: $T_{spiral}(x, y) = \exp[-i \cdot l \cdot atan2(y, x)]$, where $l \in \mathbb{Z}$ (i.e. integer) governs the periodicity of phase variation along the azimuthal direction, also referred to as the azimuthal index or the topological charge.

The SLM encodes phase information $\varphi^i$ into higher-order diffracted beams by varying the starting phase value of a horizontal, blazed-grating-like phase profile with uniform periodicity[41]. This phase profile is masked by a circular aperture centred at $(x^i, y^i)$ with the radius $r$ to match the circular profiles of the collimated beams. Consequently, the collimated, Gaussian-shaped incident beam is transformed into a centrosymmetric, closely packed array of collimated output beamlets, each with an identical circular shape and a varying intensity distribution. These beamlets emerge as higher-order diffraction patterns along the horizontal direction, with the first-order diffracted beam being the most intense (among higher-order diffracted beams) and thus selected for use. To further suppress interference fringes from the zeroth-order diffraction, beyond the diffraction order suppression system, a vertical blazed-grating-like pattern is superimposed on areas outside the simulated fibre regions, effectively redirecting the zeroth-order diffraction along the vertical direction as well. Given the Gaussian profile of the incident beam, the resulting beamlets exhibit non-uniform intensity distributions. To homogenise these intensities, additional vertical binary grating patterns[42] are applied to each simulated fibre output, modulating their intensities atop the blazed grating for phase control. This adaptive modulation reduces the intensity of the brighter beamlets more substantially, ensuring a uniform far-field

intensity across all beamlets. Importantly, this modulation affects only the intensities of the beamlets at far field, without changing the intensity distributions of the beamlets. Whilst previous studies have demonstrated that SLM can simultaneously and independently modulate both phase profiles and intensity distributions[43,44], these methods often require multiple SLMs and high pixel resolution to preserve beam quality, imposing stringent constraints. Such limitations reduce their practicality for this work, which requires the generation of SLM patterns encoding both phase and intensity information across a large number of beamlets. An example of the phase-encoding pattern displayed on the SLM is presented in Fig. 6(c). The phase calculation and the phase-and-intensity-encoded SLM pattern generation are performed on a *NVIDIA Quadro P6000* GPU-powered Windows 10 workstation. The total time required to calculate the phases for 61 beamlets, generate the corresponding pattern, and apply it to the SLM is measured to be 0.086 ± 0.004 s.

In our SLM-based, proof-of-principle experimental setup, the relative phases of all beamlets are inherently known and effectively noise-free, with phase control achieved directly by assigning phase values to spatially segmented beamlets. High-power, fibre-based CBC systems, on the other hand, face several additional complexities. Firstly, the phases of individual fibre outputs are not directly accessible and are subject to random phase noise, necessitating additional phase retrieval hardware and/or software in a real-time feedback control loop. Algorithmic approaches for phase locking (e.g. SPGD), which have been demonstrated to scale effectively to high-channel-count CBC systems[22] without requiring additional phase-sensing hardware, typically extract phase information by observing the far-field intensity distribution (or PIB measurement through a pinhole). However, since the phase control strategy demonstrated in this work enables three-dimensional beam steering by approximating the phase profile of an additional lens, the Fourier plane is effectively shifted axially after the phase profile is applied. This shift introduces new challenges for far-field, observation-based algorithms like SPGD, as it complicates the extraction of accurate phase information. Hardware-based approaches (e.g. heterodyne detection) often require additional optical components, such as microlens arrays and/or diffractive optical elements, which must be precisely aligned in multiple axes and, in some cases, custom-fabricated. These hardware additions pose substantial challenges for scalability, particularly when approximating transmissive phase-delay optical elements using a large number of channels, as discussed in Section A and Section F of the Supplementary Information. Secondly, phase control for fibre outputs is typically implemented via physical actuators such as fibre stretchers or electro-optic modulators (e.g. Lithium Niobate modulators), whose response characteristics are often nonlinear and susceptible to environmental variations. These additional complexities present major engineering challenges in scaling the proposed phase control strategies to high-power, fibre-based CBC systems. Nonetheless, we believe the phase control strategies demonstrated in this work provide a solid foundation for future efforts aimed at implementing a high-power, fibre-based CBC system with enhanced capability and flexibility.

## Data availability

Data underlying the results presented in this paper are available in https://doi.org/10.5258/SOTON/D3509.

## Code availability

Code underlying the results presented in this paper are available in https://doi.org/10.5258/SOTON/D3509.

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

## Acknowledgements

We appreciate and acknowledge funding from Engineering and Physical Sciences Research Council (EP/W028786/1, EP/T026197/1), and The Wolfson Foundation (360G-wolfson-22937).

## Author contributions

B.M., M.N.Z. and Y.X. conceptualised the work. J.G.J., M.P. and Y.X. contributed to the construction of the experimental setup. Y.X. conducted the experiments, collected the data, and performed the data analysis. B.M. and M.N.Z. supervised the project, provided direction, and secured resources. All authors contributed to the writing, editing, and review of the manuscript, and all authors approved the final version for submission.

## Competing interests

The authors declare no competing interests.
