## [Transparent Peer Review file · Communications Engineering]

Exploring five types of beam shaping using tiled-aperture coherent beam combining

Corresponding Author: Dr Yunhui Xie

Version 0:

Reviewer comments:

Reviewer #1

(Remarks to the Author)

The manuscript presents a versatile method for modulating the combined beam focus in a simulated CBC setup by leveraging phase offsets between coherently combined beams to approximate transmissive, phase-delaying optical elements. The authors demonstrate three-dimensional control over the combined beam focus, including beam steering, rotational control, and bespoke shape modulation, achieved through phase-only modulation without additional hardware. This study provides a novel perspective on CBC systems, potentially expanding their capabilities for advanced beam shaping and steering applications.

The authors have conducted excellent research, and their approach is valuable and worth publishing after a minor revision. The proposed method demonstrates significant potential; however, the novelty of the approach should be better highlighted, as the current text does not sufficiently emphasize the originality of the results. The manuscript presents strong contributions, but addressing the following points would further enhance its quality:

1. The manuscript would benefit from utilizing an original fiber array instead of modulating the beams via an SLM. Integrating a fiber array (even a 19-beam configuration would be sufficient) could enhance the practical applicability of the method and better align it with real-world high-power CBC implementations.
2. Since the study is based on modulation rather than a real high-power CBC demonstration, certain critical challenges may remain unaddressed by the proposed methods. More complex control systems are required to mitigate thermal drifts and environmental variations in high-power CBC applications. High-power operation exacerbates diffraction losses and sidelobe intensities due to the gaps between tiled beams. The authors should discuss these challenges in the text to provide a more comprehensive perspective.
3. Figure 1: Adding a 2D diagram to Figure 1b would improve clarity, as the current 3D representation looks visually appealing but does not adequately present phase differences between beams.
4. Figure 6: The figure should indicate the position of the convex second lens f_2 (where the lens is supposed to be placed) to clarify its placement in the setup. Alternatively, an additional simplified experimental concept scheme (an advanced version of Fig. 1a3) could be provided to improve understanding of the current schematic.

Overall, the paper presents significant contributions and is worthy of publication after these revisions.

Reviewer #2

(Remarks to the Author)

In the main content section, The manuscript does not provide a comprehensive comparison with relevant existing works. It is essential to clearly differentiate the work from previous studies by highlighting the specific improvements, unique features, or superior performance of your approach. However, the manuscript only briefly mentions some related works without in-depth analysis or comparison of the results. This makes it difficult for readers to understand the true value and significance of the contributions of the current work. A more thorough discussion on how the work advances the field compared to existing solutions is necessary.

1. The manuscript does not adequately discuss the extensibility of the proposed method in practical CBC system. For instance, the phase control approach relies on phase-locked loop control, which may be challenging to implement in high-power systems due to thermal effects and phase noise. The manuscript does not address how an evaluation function could be established to measure the quality of the beam focus in 3D space and how this metric could be used to achieve closed-loop control in the presence of phase noise and thermal effects in CBC systems. The manuscript would benefit from a discussion of how the proposed 3D focus control method could be integrated into a closed-loop control system. For example:

- (1) What evaluation function could be used to measure and compensate for phase errors in real-time to maintain the desired focus position and shape?
- (2) What are the potential challenges in implementing such a closed-loop system, and how could they be addressed?

2. The manuscript presents simulation results for 3D beam steering and shaping, while the analysis provided is relatively superficial and lacks depth in terms of comparing the performance of different structured beams, such as the axial depth-of-focus control capabilities of different structured beams or the achievable transverse steering angles. These parameters are essential for evaluating the practical utility of the proposed method in applications such as laser material processing, optical trapping, or directed energy systems. To enhance the scientific value of the manuscript, I recommend that the authors provide a more in-depth analysis of the beam control capabilities, including:

- (1) A comparative study of the axial depth-of-focus control for different structured beams, highlighting their respective advantages and limitations.
- (2) An analysis of the maximum achievable transverse steering angles for different beam types, along with a discussion of the factors that limit these angles (e.g., phase modulation resolution, array geometry).
- (3) A discussion of how the beam control performance varies with key parameters such as the number of fibers, fill factor, and phase noise levels.

Such an analysis would provide valuable insights into the practical capabilities and limitations of the proposed method, making the manuscript more impactful and useful for researchers specializing in high-power lasers and beam shaping.

3. The manuscript primarily focuses on simulation results to demonstrate beam shaping and steering using a tiled-aperture CBC system. However, the experimental validation is limited to the generation of vortex beams (i.e., beams carrying orbital angular momentum) and does not extend to the 3D beam control capabilities discussed in the simulations. I recommend that the authors utilize the experimental setup to verify the 3D beam steering and shaping capabilities and show the corresponding experimental results.

4. The manuscript does not clearly specify the exact figure numbers in the supplementary materials when referring to them in the main text. For example, the manuscript mentions "a more detailed analysis of the approximation ability and its impact on the power-in-bucket metric is provided in the Supplementary Information" (Page 3) and "fork-shaped interference patterns between OAM beams with different modes and a reference beam are shown in the Supplementary Information" (Page 7), but it does not provide precise references to the corresponding supplementary figures. This lack of clear referencing makes it difficult for readers to locate and correlate the supplementary materials with the main text, thereby hindering the overall readability and comprehension of the manuscript.

To improve clarity, I recommend that the authors explicitly label and reference each supplementary figure, or section in the main text (e.g., "see Supplementary Figure S1"). This will greatly enhance the reader's ability to follow the discussion and fully appreciate the supplementary data provided.

5. The manuscript does not provide detailed information on the dimensions of the array used in the simulations, such as the shape of each beamlet and diameter of aperture size. This omission makes it challenging to assess the scalability of the proposed method and, more importantly, hinders the reproducibility of the results.

6. I noticed that in the principal focal plane f_1 position depicted in Fig. 3 and Fig. 4, there are more than six side lobes surrounding the main lobe of the simulated results. However, according to the parameters provided by the authors, such as the fill factor and focal lens, it is theoretically expected that a hexagonally close-packed array should have only six side lobes around the main lobe. OR whether the authors do not give some additional calculation parameters? Therefore, the accuracy of the authors' simulation calculations may be subject to further verification.

Reviewer #3

(Remarks to the Author)

This manuscript presents a comprehensive and insightful study on the capabilities of coherent beam combining (CBC) systems not only for power scaling but also for advanced beam shaping and steering. Through both theoretical modeling and experimental validation, the authors demonstrate that phase-only control in a tiled-aperture CBC configuration can approximate a variety of optical components (spherical and axicon lenses, spiral phase plates), allowing for sophisticated beam manipulation in three dimensions and in multiple modes (Bessel, OAM). The work is well-structured, technically rigorous, and timely, with clear relevance to high-power laser systems and adaptive optics.

- While the use of an SLM is sufficient for proof-of-principle, a brief discussion on how the proposed phase control strategies would scale to actual high-power fibre CBC hardware would strengthen the paper's practical relevance.

- Although the authors mention "power-in-bucket" in the supplementary information, integrating more quantitative beam quality metrics (e.g., M^2 , Strehl ratio) in the main text would aid in evaluating the fidelity of the shaped beams.

- A few minor grammatical errors and inconsistent figure referencing (e.g., "Figures 5(c4), Figures 5(d4), Figures 5(c3)") should be polished for clarity.

Addressing the points above will enhance its clarity and practical impact.

Version 1:

Reviewer comments:

Reviewer #1

(Remarks to the Author)

The authors have sufficiently addressed the questions and comments; the paper merits acceptance.

Reviewer #3

(Remarks to the Author)

The authors have addressed my previous concerns in the revised version. The clarifications and additional revisions have improved the manuscript. I have no further objections and recommend acceptance of the paper.

27 May 2025

Dear Editors and Reviewers,

On behalf of all the authors, I would like to express our sincere gratitude for your valuable time and constructive comments on our manuscript. Your detailed and thoughtful suggestions have greatly assisted us in enhancing both the clarity and scientific rigor of our work.

In the following document, we provide a comprehensive, point-by-point response (colour-coded in **blue**) addressing each of the reviewers' comments. For clarity and ease of comparison, we have included excerpts from our original manuscript text colour-coded in **green**, followed by the revised text colour-coded in **red**, clearly highlighting the modifications and improvements made based on your comments and suggestions.

We have addressed each comment and suggestion raised, with which we fully agree, and we believe that these revisions have substantially improved the manuscript, ensuring it now fully meets the high standards required for publication. Thank you again for your critical insights and support throughout this process.

Yours sincerely,

Dr. Yunhui Xie

Research Fellow

Optoelectronics Research Centre

University of Southampton

Southampton, SO17 1BJ

Reviewer #1 (Remarks to the Author):

The manuscript presents a versatile method for modulating the combined beam focus in a simulated CBC setup by leveraging phase offsets between coherently combined beams to approximate transmissive, phase-delaying optical elements. The authors demonstrate three-dimensional control over the combined beam focus, including beam steering, rotational control, and bespoke shape modulation, achieved through phase-only modulation without additional hardware. This study provides a novel perspective on CBC systems, potentially expanding their capabilities for advanced beam shaping and steering applications.

The authors have conducted excellent research, and their approach is valuable and worth publishing after a minor revision. The proposed method demonstrates significant potential; however, the novelty of the approach should be better highlighted, as the current text does not sufficiently emphasize the originality of the results. The manuscript presents strong contributions, but addressing the following points would further enhance its quality:

We sincerely thank the reviewer for their thoughtful and insightful evaluation of our manuscript. We fully agree with the reviewer's observation that the novelty and significance of our contributions to advancing the current state of the art could have been more clearly articulated in the original submission. To address this important point, we have revised Section 2 (Page 1, starting at Line 31) to more clearly and explicitly highlight the key innovations and their relevance to the field.

“A key challenge in devising a CBC system is the suppression of fluctuating phase noise to maintain mutual coherence between fibre outputs

Beyond power scaling capabilities, CBC systems have also been extensively studied for their potential as phased arrays, enabling further control over the combined beam focus such as agile beam steering and beam shaping. Commercially, *Civan Lasers* has demonstrated advanced laser welding applications utilising dynamic beam steering and shaping capabilities via CBC systems. Although the technical details of these capabilities have not been fully disclosed, it is stated that they are achieved by treating the CBC system as an Optical Phased Array (OPA). The existing literature addresses several innovative extensions of CBC functionalities. *Chang et al.* reported on modulating the phase profile of a CBC system to approximate a spaced dual-lens configuration, thereby achieving axial steering of the combined beam focus. *Jabczyński* demonstrated the generation of Bessel beams using CBC, whilst various other studies have explored the generation of beams carrying Orbital Angular Momentum (OAM) in CBC system.

In this work, we propose a novel theoretical framework that leverages phase-only control of relative phases between CBC channels to approximate the phase profiles of transmissive, phase-delaying optical elements, such as lenses, axicons, and spiral phase plates. This approach allows the functionalities of these optical elements to be

approximated by a CBC system functioning as a phased array, without requiring additional hardware. Our investigation begins with the phase-only approximation of a lens in a contacting dual-lens configuration, through which we demonstrate simultaneous steering of the combined beam focus along both axial (i.e., z) and transverse (i.e., x and y) directions. This is accomplished by adjusting the effective focal length and applying lens decentring aberrations to a “virtual” lens created by treating the CBC system as a phase array. This SLM-based “virtual” lens is used in a combination with a conventional lens to approximate a contacting dual-lens configuration. Furthermore, we extend this concept to demonstrate that three-dimensional steering capabilities are not limited solely to coherently combined focal spots but are also applicable to various structured beam profiles, including Bessel-like beams and beams with defined OAM states. Finally, we present a video demonstration showcasing simultaneous phase-only control of the combined beam focus, illustrating capabilities including: (1) varying the spatial position along axial (i.e., z) and transverse (i.e., x and y) directions, (2) dynamic beam shaping through rotation of the transverse intensity distribution, and (3) controllable orbital angular momentum, characterised by adjusting the rotation rate of the helical phase front with propagation.”

We hope that these changes have clarified and highlighted the contributions and novel aspects of our research more explicitly.

1. The manuscript would benefit from utilizing an original fiber array instead of modulating the beams via an SLM. Integrating a fiber array (even a 19-beam configuration would be sufficient) could enhance the practical applicability of the method and better align it with real-world high-power CBC implementations.

We fully agree with the reviewer’s valuable comment regarding the significant benefit of experimentally demonstrating our proposed method using a real-world fibre-based setup, rather than employing an SLM for beam modulation. Such a demonstration would undeniably enhance the practical relevance and validity of our theoretical framework.

To address transparently the potential differences between our current SLM-based experimental results and those achievable in real-world fibre laser setups, we have added the following discussion to Section 2 (Page 3, starting at Line 78, a portion of the following revised text has been replaced with “...” as it pertains to a different comment, which will be addressed in the subsequent response):

“Details of the experimental setup, including the simultaneous control of the phase and intensity of each simulated fibre output, and the phase offset calculations for approximating the second lens f_2 with varying focal lengths for axial combined beam focus steering, are provided in the Methods section. Although the SLM approach, in theory, should be able to simulate the combined output of an ideal tiled-aperture CBC system, real-world, fibre-based, high-power CBC systems rarely achieve such idealised outputs due to a range of practical complications.... Together, these complications highlight the gap between the idealised SLM-based demonstration and a practical fibre-based, high-power CBC system, underscoring the need for further development to translate the proof-of-principle concept into a fibre-based experimental demonstration, which we aim to pursue in future work.”

Moreover, to ensure complete transparency and reproducibility, we have deposited all relevant data and code in an open-access research repository managed by our institutional library. The dataset is associated with a DOI, which is provided in the Data Availability Statement. However, in accordance with the library’s open-access data policy, the DOI short link will only be activated and made publicly accessible upon acceptance of the manuscript, as the accepted version of the manuscript must be uploaded alongside the dataset. As such, the dataset may not be accessible during peer review. If the reviewer would like to access the dataset at this stage, we would be happy to discuss with the library to generate a temporary access link.

2. Since the study is based on modulation rather than a real high-power CBC demonstration, certain critical challenges may remain unaddressed by the proposed methods. More complex control systems are required to mitigate thermal drifts and environmental variations in high-power CBC applications. High-power operation exacerbates diffraction losses and sidelobe intensities due to the gaps between tiled beams. The authors should discuss these challenges in the text to provide a more comprehensive perspective.

We fully agree with the reviewer that the discrepancies between the demonstrated SLM-based approach and a high-power, fibre-based CBC system should be more explicitly addressed. In response to this and the earlier comment, we have now included a detailed discussion highlighting these discrepancies in Section 2 (Page 3, starting at Line 78, a portion of the following revised text has been replaced with “...” as it pertains to a different comment, which was addressed in the previous response):

“Details of the experimental setup, including the simultaneous control of the phase and intensity of each simulated fibre output, and the phase offset calculations for approximating the second lens f_2 with varying focal lengths for axial combined beam focus steering, are provided in the Methods section. ...due to a range of practical complications. For instance, misalignments arising from tip-tilt and installation errors introduce wavefront aberrations, which must be accounted for during phase profile approximation. Moreover, high-power operation typically requires more complex system configurations, often involving additional amplification stages and active cooling components. These additions pose substantial challenges for managing thermal and environmental variations, which could compromise the performance of both phase control hardware and software. Lastly, the SLM-based approach implicitly assumes that beamlets remain perfectly collimated (i.e., diffraction-induced beam divergence and associated sidelobe growth can be neglected), whereas in a fibre-based, high-power, tiled-aperture CBC system each fibre output will undergo finite diffraction, leading to extra losses and increased sidelobe intensities. Together, ...”

3. Figure 1: Adding a 2D diagram to Figure 1b would improve clarity, as the current 3D representation looks visually appealing but does not adequately present phase differences between beams.

We thank the reviewer for pointing out the missing information in Figure 1b. In response, we have updated the figure to include a 2D side view of the phase delay for clarity:

We have also updated the subfigure labelling in Figure 1 to improve clarity. The revised caption for Figure 1 is provided below (Page 2, starting at Line 47):

“(b1), The phase delays imposed by a lens can be described by a parabolic function; a corresponding side view is shown in (b2). (c1), Similarly, inserting a Fresnel lens shifts the focal plane and the combined beam axially (c2), The phases of the beams are sampled from the phase offsets imposed by a Fresnel lens via taking the phase values at the geometric centres of the beam locations.”

The revised main text is provided below (Page 2, starting at Line 63):

“If the second lens f_2 is a simple spherical lens imposing quadratic phase delays, these delays can be approximated by fitting the phase profiles of the beams to a parabolic function, as illustrated in Figure 1(b1) and Figure 1(b2). Alternatively, if the second lens f_2 is a Fresnel lens, the phase offsets can be sampled from the phase profile of the Fresnel lens, by taking the phase values at the geometric centres of the beams as illustrated in Figure 1(c1) and Figure 1(c2). The physical meanings of both approaches are almost identical, as a Fresnel lens approximates a spherical lens by segmenting the continuous parabolic phase profile into discrete steps (i.e., wrapping the phase), and both approaches aim to impose a quadratic phase delay across the aperture, either continuously as shown in Figure 1(c1) (i.e., spherical lens) or discretely as shown in Figure 1(c2) (i.e., Fresnel lens).”

4. Figure 6: The figure should indicate the position of the convex second lens f_2 (where the lens is supposed to be placed) to clarify its placement in the setup. Alternatively, an additional simplified experimental concept scheme (an advanced version of Fig. 1a3) could be provided to improve understanding of the current schematic.

We thank the reviewer for pointing out that the physical lens f_1 and the virtual lens f_2 are not explicitly indicated in Figure 6(a). As suggested, we have added a schematic to Figure 6(a) to explicitly identify both the physical and virtual lenses:

Overall, the paper presents significant contributions and is worthy of publication after these revisions.

We sincerely thank the reviewer once again for their thoughtful comments and encouraging feedback. We hope that the revisions made in response will meet the high standards required for publication.

Reviewer #2 (Remarks to the Author):

In the main content section, the manuscript does not provide a comprehensive comparison with relevant existing works. It is essential to clearly differentiate the work from previous studies by highlighting the specific improvements, unique features, or superior performance of your approach. However, the manuscript only briefly mentions some related works without in-depth analysis or comparison of the results. This makes it difficult for readers to understand the true value and significance of the contributions of the current work. A more thorough discussion on how the work advances the field compared to existing solutions is necessary.

We would like to express our sincere gratitude to the reviewer for the thoughtful and constructive feedback. We fully agree that a more explicit and thorough comparison with relevant existing works is crucial for clarifying the novelty and significance of our contributions. We have expanded Section 2 (Page 1, starting at Line 31) to better highlight how our approach differs from, and improves upon, prior studies. To avoid replication, we kindly refer the reviewer to our response to Reviewer 1, Comment 1, where the revised manuscript text is provided.

1. The manuscript does not adequately discuss the extensibility of the proposed method in practical CBC system. For instance, the phase control approach relies on phase-locked loop control, which may be challenging to implement in high-power systems due to thermal effects and phase noise. The manuscript does not address how an evaluation function could be established to measure the quality of the beam focus in 3D space and how this metric could be used to achieve closed-loop control in the presence of phase noise and thermal effects in CBC systems. The manuscript would benefit from a discussion of how the proposed 3D focus control method could be integrated into a closed-loop control system. For example:

(1) What evaluation function could be used to measure and compensate for phase errors in real-time to maintain the desired focus position and shape?

(2) What are the potential challenges in implementing such a closed-loop system, and how could they be addressed?

We thank the reviewer for the insightful and constructive comment regarding the extensibility of the demonstrated phase control strategies in a fibre-based, high-power CBC system. We fully agree that the manuscript would benefit from a more in-depth discussion on how the demonstrated phase control strategies could be integrated into a practical, closed-loop control system that accounts for phase noise, thermal effects, and other high-power-related challenges. In response, we propose the following additions to the manuscript:

To Section 2 (Page 3, starting at Line 78, a portion of the following revised text has been replaced with “...” as it pertains to a different comment, which was addressed in a previous response), we would like to include the following discussion to clarify the high-power-related challenges in high-power CBC systems:

“Details of the experimental setup, including the simultaneous control of the phase and intensity of each simulated fibre output, and the phase offset calculations for approximating the second lens f_2 with varying focal lengths for axial combined beam focus steering, are provided in the Methods section. . . . Moreover, high-power operation typically requires more complex system configurations, often involving additional amplification stages and active cooling components. These additions pose substantial challenges for managing thermal and environmental variations, which could compromise the performance of both phase control hardware and software.”

To Section 5 (Page 9, starting at Line 311), we would like to include the following discussion to address the reviewer’s comments on the potential challenges in implementing the demonstrated phase control strategies in a fibre-based, high-power CBC system:

“The phase calculation and the phase-and-intensity-encoded SLM pattern generation are performed on a *NVIDIA* Quadro P6000 GPU powered Windows 10 workstation. The total time required to calculate the phases for 61 beamlets, generate the corresponding pattern, and apply it to the SLM is measured to be 0.086 ± 0.004 seconds.

In our SLM-based, proof-of-principle experimental setup, the relative phases of all beamlets are inherently known and effectively noise-free, with phase control achieved directly by assigning phase values to spatially segmented beamlets. High-power, fibre-based CBC systems, on the other hand, face several additional complexities. Firstly, the phases of individual fibre outputs are not directly accessible and are subject to random phase noise, necessitating additional phase retrieval hardware and/or software in a real-time feedback control loop. Algorithmic approaches for phase locking (e.g., SPGD), which has been demonstrated to scale effectively to high-channel-count CBC systems without requiring additional phase-sensing hardware, typically extract phase information by observing the far-field intensity distribution (or PIB measurement through a pinhole). However, since the phase control strategy demonstrated in this work enables three-dimensional beam steering by approximating the phase profile of an additional lens, the Fourier plane is effectively shifted axially after the phase profile is applied. This shift introduces new challenges for far-field, observation-based algorithms like SPGD, as it complicates the extraction of accurate phase information. Hardware-based approaches (e.g., heterodyne detection) often require additional optical components, such as microlens arrays and/or diffractive optical elements, which must be precisely aligned in multiple axes and, in some cases, custom-fabricated. These hardware additions pose significant challenges for scalability, particularly when approximating transmissive phase-delay optical elements using a large number of channels, as discussed in Section A and Section F of the Supplementary Information. Secondly, phase control for fibre outputs is typically implemented via physical actuators such as fibre stretchers or electro-optic modulators (e.g., Lithium Niobate modulators), whose response characteristics are often nonlinear and susceptible to environmental variations. These additional complexities

present significant engineering challenges in scaling the proposed phase control strategies to high-power, fibre-based CBC systems. Nonetheless, we believe the phase control strategies demonstrated in this work provide a solid foundation for future efforts aimed at implementing a high-power, fibre-based CBC system with enhanced capability and flexibility.”

In response to the reviewer’s comment regarding the evaluation function, we would like to present the following simulation results, which are part of a separate line of ongoing work focused on addressing the algorithmic phase control problem in tiled-aperture CBC systems.

Figure (A1) and Figure (A2) show the simulated intensity distributions at the principal focal plane and 5 cm before the focal plane, respectively, using the same phase profile as shown in Figure (A3) and described in the manuscript. Figure (B) shows an SPGD-based optimisation process in which, instead of maximising a metric such as PIB, which increases as phase differences between channels decrease, a different performance metric is used that measures the dissimilarity between the observed intensity distribution I_{obs} and a target distribution I_{target} . Specifically, the objective function is defined using the L^1 -norm:

$$J(I_{obs}, I_{target}) = \sum_x \sum_y |I_{obs}(x, y) - I_{target}(x, y)|$$

Minimising this objective function (i.e., updating the phase $\varphi^{t+1} \leftarrow \varphi^t - \delta J \cdot \delta \varphi^t$, rather than the more conventional $\varphi^{t+1} \leftarrow \varphi^t + \delta J \cdot \delta \varphi^t$ which is used to maximised PIB) allows the beamlets to constructively interfere 5 cm before the principal focal plane, even though only the intensity at the principal focal plane is observed during optimisation. Figure (B) shows an instance of this optimisation starting from a random initial phase configuration. Figures (C1), (C2), and (C3) show the intensity distributions at the focal plane, 5 cm before the focal plane, and the corresponding phase profile at the first iteration step. Figures (D1), (D2), and (D3) show the same at an intermediate step. Figures (E1), (E2), and (E3) show the results at the final optimisation step.

We note that there are practical issues associated with this modified SPGD approach. For example:

- (1) the injectivity of the objective function J with respect to beam positions along the axial direction still need to be rigorously evaluated.
- (2) the perturbation term $\delta \varphi$ may require dynamic scaling to account for small phase differences during convergence.

Nonetheless, the results suggest that this approach is potentially a viable option for enabling the phase control strategies presented in the manuscript, particularly when closed-loop control is required. This line of work, which targets general algorithmic phase locking for bespoke, multi-axis beam shaping in CBC systems, is ongoing. We believe that including a full treatment of these discussions would deviate too far from the main focus of this manuscript. Therefore, we have opted to present these results only as a brief illustration and will report the full framework in a separate, future publication.

2. The manuscript presents simulation results for 3D beam steering and shaping, while the analysis provided is relatively superficial and lacks depth in terms of comparing the performance of different structured beams, such as the axial depth-of-focus control capabilities of different structured beams or the achievable transverse steering angles. These parameters are essential for evaluating the practical utility of the proposed method in applications such as laser material processing, optical trapping, or directed

energy systems. To enhance the scientific value of the manuscript, I recommend that the authors provide a more in-depth analysis of the beam control capabilities, including:

- (1) A comparative study of the axial depth-of-focus control for different structured beams, highlighting their respective advantages and limitations.
- (2) An analysis of the maximum achievable transverse steering angles for different beam types, along with a discussion of the factors that limit these angles (e.g., phase modulation resolution, array geometry).
- (3) A discussion of how the beam control performance varies with key parameters such as the number of fibers, fill factor, and phase noise levels.

Such an analysis would provide valuable insights into the practical capabilities and limitations of the proposed method, making the manuscript more impactful and useful for researchers specializing in high-power lasers and beam shaping.

We thank the reviewer for bringing our attention to the lack of discussion regarding the tunability of the structured beams. **We would first like to clarify that all results presented in the manuscript were obtained experimentally using the setup described in Section 5. No simulated data were presented or discussed in any part of the manuscript.**

In response to the reviewer's comment, we have added a new section to the Supplementary Information demonstrating the axial tuneable of LG_{01} , LG_{02} , and LG_{03} OAM beams.

"F. Axial and transverse tuneability for LG_{01} , LG_{02} , and LG_{03} OAM beams via approximating spiral phase plates and lenses simultaneously

Fig. S6. Intensity distributions along the axial direction of three different OAM modes generated by arrays of 19, 61, and 127 beamlets with radii of 84, 42, and 28 pixels, respectively (so that all aperture sizes are approximately equal). Rows correspond to increasing array size (from 19 to 61 to 127 beamlets, top to bottom), and columns correspond to the LG_{01} , LG_{02} , and LG_{03} modes (left to right).

By approximating the combined phase profile of a spiral phase plate and a lens, the axial position of the focal point of an OAM-carrying beam can be controlled, as discussed in Section 3.4. Figure S6 presents additional examples of the axial intensity distributions for the LG_{01} , LG_{02} , and LG_{03} modes, each approximated using arrays of 19, 61, and 127 beamlets with approximately equal aperture sizes. As shown in Figure S6, increasing the number of beamlets, whilst keeping the aperture size approximately the same, extends the axial steering range of the OAM beam. In addition, as (the absolute value of) the topological charge l increases, the axial steering range of the OAM beam becomes more limited. This observation is consistent with the discussion presented in Section A, which describes how increased phase variation (arising from a reduced focal length of the approximated lens) prevents the piston phase of a beamlet from accurately approximating the continuous phase profile within its own spatial extent. To maintain a sufficient approximation under stronger phase variation, a smaller beamlet radius is required. Different from the discussion made in Section A, where the increased phase variation arises solely from modulating the focal length of approximated lens, the shown OAM beams involve additional phase variations in the azimuthal directions due to the higher topological charges l . Moreover, increasing the radial index p would further intensify phase variation, leading to similar limitations. However, for simplicity, the effects of increasing the radial index p are not included here. The intensity distributions along the transverse direction at axial positions of -10.0 , -7.5 , -5.0 , -2.5 , 0 , 2.5 , 5.0 , 7.5 , and 10.0 cm are shown in Figure S7.

Fig. S7. Intensity distributions along the transverse direction of beams carrying the LG_{01} , LG_{02} , and LG_{03} modes, each approximated by arrays of 19, 61, and 127 beamlets. Intensity distributions are shown at axial positions of -10.0 , -7.5 , -5.0 , -2.5 , 0 , 2.5 , 5.0 , 7.5 , and 10.0 cm.

The transverse tuneability for the beams carrying LG_{01} mode is demonstrated in Figure S8, where, at progressively decreasing axial positions, an increasing lens decentring is applied. This allows transverse steering of the beams carrying LG_{01} mode along the x-axis at multiple axial positions. As detailed in Section 3.2 and Section 3.3, the effective focal length of the second lens, f_2 , tends to infinity (i.e., the lens behaves as a plane-parallel plate) when the combined beam focus is axially steered to the principal focus of lens f_1 . Consequently, decentring lens f_2 at this principal focus results in no transverse steering of the combined beam focus, as observed in the first row of Figure S8. Here, increased lens decentring progressively distorts the beam profile without shifting its position along the x-axis. Furthermore, Figure S8 also demonstrates that as the beam is steered axially away from the principal focus of f_1 , identical amounts of lens decentring result in progressively larger transverse displacements of the combined beam focuses. This observation aligns with the results discussed in Section 3.2 and Section 3.3. Conceptually, lens decentring can be interpreted as introducing an angular tilt to the beam relative to the principal focus of f_1 . The maximum achievable tilt angle (without visually distorting the intensity

distribution) for the beam carrying LG_{01} mode and the corresponding beamlet array characterised in Figure S8 is measured to be approximately 0.190° . A more advanced method of combined beam steering, involving the simultaneous approximation of the phase profiles of both a decentred lens and a wedge prism, has been demonstrated previously in Section B.

Fig. S8. Intensity distributions along the transverse direction of beams carrying the LG_{01} mode, approximated by an array of 127 beamlets (radii of 28 pixels). Each column shows the intensity distributions at a decreasing axial distance (0.0, -2.5, -5.0, -7.5 and -10.0 cm from top to bottom), whilst each row shows the intensity distributions at the same position with an increasing lens decentring (from 0.00 mm to 2.92 mm, left to right).

Figure S9 demonstrates the transverse tuneability for the beams carrying LG_{01} , LG_{02} , and LG_{03} modes at an axial position -7.5 cm from the principal focus of lens f_1 , each mode approximated by the same beamlet array (127 beamlets with 28-pixel radii). It is evident that as the topological charge l increases, transverse tuneability decreases. Specifically, at higher lens decentring values, the resultant intensity distribution progressively deviates further from the ideal ring-like intensity distribution (from OAM mode). This decrease in transverse tuneability with increasing topological charge l is similar to the previously observed decrease in axial tuneability; this occurs because higher topological charges l introduce greater phase variations, making piston phase less effective at approximating the continuous phase profile within each beamlet's spatial extent. The maximum achievable tilt angles (without significant visual distortion of the ring-like intensity distribution) measured for the beams carrying LG_{01} , LG_{02} , and LG_{03} modes in Figure S9 are approximately 0.190° , 0.159° , and 0.096° , respectively.

Fig. S9. Intensity distributions along the transverse direction of beams carrying the LG_{01} , LG_{02} , and LG_{03} mode (first, second, and third row, respectively), each approximated by an array 127 beamlets (radii of 28 pixels). All intensity distributions are presented at the axial position of -7.5 cm from the principal focus of the lens f_1 , and each row illustrates the effect of increasing lens decentring, progressing from 0.00 mm to 2.92 mm (left to right).

Another critical factor affecting the transverse tuneability of beams carrying OAM modes is the aperture size of the beamlet array. Significant lens decentring can displace the phase singularity (i.e., the vortex core) beyond the restricted aperture of the beamlet array, leaving only residual fringe-like phase patterns at axial positions away from the principal focus of lens f_1 . This condition results in a spot-like intensity distribution with transverse displacement, effectively approximating a wedge prism. Figure S10 demonstrates the transverse tuneability of beams carrying the LG_{01} mode, approximated by beamlet arrays of progressively larger aperture sizes achieved by increasing the number of beamlets (from 19 in the first row to 61 in the third row, and 127 in the fifth row), whilst maintaining constant beamlet radii. It can be observed that with smaller aperture sizes, the characteristic ring-like shape of the LG_{01} mode deteriorates into a single spot as the phase singularity exits the limited aperture. Conversely, expanding the aperture size by increasing the number of beamlets improves transverse tuneability by maintaining the phase singularity within the aperture, even at higher lens decentring values.

Fig. S10. Intensity distributions along the transverse direction of beam carrying the LG_{01} mode, approximated by arrays of 19, 61, and 127 beamlets (first, third, and fifth rows, respectively); each beamlet has an identical radius of 28 pixels, leading to progressively increasing aperture sizes. The corresponding target phase profiles, which the beamlet arrays aim to approximate for simultaneous axial and transverse steering, are shown in the second, fourth, and sixth rows for the 19, 61, and 127 beamlet arrays, respectively, with outlines of the beam arrays overlaid on the phase profiles. All intensity distributions are presented at the axial position of -7.5 cm from the principal focus of the lens f_1 , and each row illustrates the effect of increasing lens decentring, progressing from 0.00 mm to 2.92 mm (left to right).

From the discussions and results presented above, it can be concluded that the beam steering capability of the proposed phase control strategy is fundamentally governed by the physical characteristics of the beamlet array. Specifically, arrays with a higher number of smaller beamlets are preferable, as they facilitate combined beam focus steering across a larger space and support higher-order modes. However, devising a compact, high-power, and high-channel-count CBC system presents substantial engineering challenges. Overcoming these obstacles, including thermal management and phase noise suppression, will be the focus of our future research efforts.”

3. The manuscript primarily focuses on simulation results to demonstrate beam shaping and steering using a tiled-aperture CBC system. However, the experimental validation is limited to the generation of vortex beams (i.e., beams carrying orbital angular momentum) and does not extend to the 3D beam control capabilities discussed in the simulations. I recommend that the authors utilize the experimental setup to verify the 3D beam steering and shaping capabilities and show the corresponding experimental results.

We thank the reviewer for raising the concern regarding the experimental validation of the proposed phase control methods. **We would like to clarify that all results, including all figures and discussions, presented in the manuscript were obtained solely from the experimental setup; no results from computational simulations were included, referenced, or discussed in any part of the manuscript.** To ensure full transparency and reproducibility, we have deposited all relevant data and code associated with this work, enabling third parties to faithfully reproduce and verify all figures in an open-access research repository managed by our institutional library. A DOI short link to this dataset is provided in the Data Availability Statement. However, we note that, in accordance with the policy of the institutional repository managing our data, the DOI link can only be activated upon manuscript acceptance, as the accepted version of the manuscript must be uploaded and linked to the dataset. Should the reviewer wish to access the dataset during the peer review process, we would be more than happy to coordinate with our library team to generate an alternative temporary access link for the reviewer's convenience.

We would like to include the following statement (Page 2, starting at Line 76) in the manuscript to explicitly clarify the source of the data and measurement approach:

“To observe the spatial intensity distribution after the lens f_1 , a camera was mounted on a motorised translation stage with a 45 cm travel range, allowing it to be moved along the beam propagation axis (i.e., axial direction). The camera directly observes the intensity distribution in the transverse plane (i.e., x–y plane), whilst the intensity distribution along the axial direction is reconstructed by stitching together multiple transverse intensity distributions captured at discrete axial positions. All figures and discussions presented in this manuscript are based solely on experimental results, and all relevant raw data and code have been open-sourced (see Data availability statement and Code availability statement).”

4. The manuscript does not clearly specify the exact figure numbers in the supplementary materials when referring to them in the main text. For example, the manuscript mentions "a more detailed analysis of the approximation ability and its impact on the power-in-bucket metric is provided in the Supplementary Information" (Page 3) and "fork-shaped interference patterns between OAM beams with different modes and a reference beam are shown in the Supplementary Information" (Page 7), but it does not provide precise references to the corresponding supplementary figures. This lack of clear referencing makes it difficult for readers to locate and correlate the

supplementary materials with the main text, thereby hindering the overall readability and comprehension of the manuscript.

To improve clarity, I recommend that the authors explicitly label and reference each supplementary figure, or section in the main text (e.g., "see Supplementary Figure S1"). This will greatly enhance the reader's ability to follow the discussion and fully appreciate the supplementary data provided.

We would like to express our sincere gratitude to the reviewer for highlighting the insufficient referencing of the figures in the Supplementary Information. In response, we have revised the manuscript to more clearly and appropriately cite the relevant supplementary figures, as detailed below:

Page 3, starting at Line 108: "A more detailed analysis of the approximation ability and its impact on the power-in-bucket metric is provided in the Supplementary Information Section A and Figure S1."

Page 4, starting at Line 144: "An example provided in the Supplementary Information Section B and Figure S2 demonstrates the simultaneous application of these two approaches in opposing tilt directions, which effectively cancels the beam tilting whilst maintaining displacement of the combined beam focus relative to the principal focal point of f_1 along the transverse directions."

Page 5, starting at Line 170: "A further example, demonstrating a shaped beam focus with a different set of phase offsets, is provided in the Supplementary Information Section C and Figure S3."

Page 7, starting at Line 215: "Furthermore, the superimposition of a spiral phase plate on an axicon lens (i.e., diffractive spiral axicon) can enable the generation of a vortex beam, as detailed in the Supplementary Information Section D and Figure S4."

Page 7, starting at Line 216: "The Supplementary Information also includes fork-shaped interference patterns (see Supplementary Information Section E and Figure S5), which determine topological charges of OAM beams, between combined beam focuses carrying different OAM modes and a reference beam."

Page 7, starting at Line 261: "The fork-shaped interference patterns between OAM beams with different modes and the reference beam are shown in the Supplementary Information Section E and Figure S5."

Page 8, starting at Line 268: "In an actual fibre CBC setup, the radii of collimated fibre outputs could be larger than the demonstrated values here, thereby limiting the minimum focal length of the approximated second lens f_2 as detailed in the Supplementary Information Section A."

5. The manuscript does not provide detailed information on the dimensions of the array used in the simulations, such as the shape of each beamlet and diameter of aperture

size. This omission makes it challenging to assess the scalability of the proposed method and, more importantly, hinders the reproducibility of the results.

We thank the reviewer for raising this important point. We would like to again clarify that the results presented in the manuscript are not derived from computational simulations. The shape and diameter of the beamlets used in the system are described in detail in Section 5 (Line 268); however, we agree that this key information should be stated more clearly earlier in the manuscript to improve clarity. To address this, we have revised the manuscript (Page 2, starting at Line 63) as follows to ensure the beam characteristics are explicitly described at an earlier stage:

“A Spatial Light Modulator (SLM) was employed to modulate a continuous-wave laser from an intensity-stabilised Helium-Neon source, simulating the collimated outputs of a hexagonally close-packed array of 61 fibres (approximated top-hat shaped with beam radii of 0.41 mm, approximately 76.3% fill factor …).”

We are grateful to the reviewer for drawing our attention to this oversight, and we hope the revision satisfactorily addresses the concern regarding the transparency and reproducibility of the beam parameters.

6. I noticed that in the principal focal plane $f1$ position depicted in Fig. 3 and Fig. 4, there are more than six side lobes surrounding the main lobe of the simulated results. However, according to the parameters provided by the authors, such as the fill factor and focal lens, it is theoretically expected that a hexagonally close-packed array should have only six side lobes around the main lobe. OR whether the authors do not give some additional calculation parameters? Therefore, the accuracy of the authors' simulation calculations may be subject to further verification.

We appreciate the reviewer's careful observation regarding the number of side lobes in the experimental results presented in Figure 3 and Figure 4. We would like to again clarify that the results shown are based on experimental measurements using a SLM, rather than computational simulations.

The presence of more than six visible side lobes in the focal plane is consistent with theoretical expectations when a lower fill factor is considered (as lower fill factor configuration distributes more power to the side lobes, thereby making them more visible, Zhou, P., Liu, Z., Xu, X., Chen, Z., & Wang, X. (2009). Beam quality factor for coherently combined fiber laser beams. *Optics & Laser Technology*, 41(3), 268-271). In our experiments, the fill factor was 76.3%, which contrasts with the >90% fill factors commonly reported in other literature. A higher fill factor would indeed produce a clearer hexagonal pattern with six dominant side lobes, but such configurations are currently impractical for our intended high-power, fibre-based CBC systems.

Through discussions with our collaborative research groups and industrial partners who are working toward scalable high-power, fibre-based implementations, we concluded that achieving a very high fill factor is not currently feasible for a number of practical reasons. A primary challenge lies in managing heat dissipation and ensuring

adequate cooling, particularly within the CBC head (e.g., micro-lens array). As a result, the fill factor used in our experiments reflects realistic design constraints for high-power operation, and we chose not to adopt an idealised or unachievable configuration in the manuscript.

For comparison, we present below the in-phase intensity distributions at focus, as observed using our experimental setup: (1) with a 90.7% fill factor, representing the theoretical maximum, and (2) with a 76.3% fill factor, corresponding to the practical configuration used in our experiments:

It can be observed that side lobes become more visible at lower fill factors. Due to the limited dynamic range of our CMOS camera, low-intensity features are not captured with high clarity. To better illustrate this effect, we have also conducted a simulation using our simulation environment, developed for a separate line of work (we would like to reiterate that all results presented in the manuscript were obtained experimentally, and the simulation environment referenced here was developed primarily for addressing the algorithmic phase control problem in a separate line of work, and none of its simulation results are included in the manuscript):

High fill ratio, far field

low fill ratio, far field

In this simulation, all beamlets are assumed to have Gaussian profiles, whereas in the experimental setup described in the manuscript, the beamlets are effectively approximated as top-hat profiles. As a result, the intensity distributions appear slightly different.

To enhance clarity in the manuscript, we have added the following statement (Page 2, starting at Line 63) to the manuscript:

“A Spatial Light Modulator (SLM) was employed to modulate a continuous-wave laser from an intensity-stabilised Helium-Neon source, simulating the collimated outputs of a hexagonally close-packed array of 61 fibres (…approximately 76.3% fill factor). Due to this relatively low fill factor, side lobes may appear more prominently.”

We thank the reviewer once again for the insightful comments. We have carefully addressed each point and made substantial revisions, and we hope that the updated manuscript now meets the high standards for publication.

Reviewer #3 (Remarks to the Author):

This manuscript presents a comprehensive and insightful study on the capabilities of coherent beam combining (CBC) systems not only for power scaling but also for advanced beam shaping and steering. Through both theoretical modeling and experimental validation, the authors demonstrate that phase-only control in a tiled-aperture CBC configuration can approximate a variety of optical components (spherical and axicon lenses, spiral phase plates), allowing for sophisticated beam manipulation in three dimensions and in multiple modes (Bessel, OAM). The work is well-structured, technically rigorous, and timely, with clear relevance to high-power laser systems and adaptive optics.

We sincerely thank the reviewer for the thoughtful and constructive feedback. We have carefully considered each of the comments and provide detailed, point-by-point responses below. We hope that the revisions made in response to these comments have improved the clarity, rigour, and overall quality of the manuscript, bringing it in line with the high standards expected for publication.

- While the use of an SLM is sufficient for proof-of-principle, a brief discussion on how the proposed phase control strategies would scale to actual high-power fibre CBC hardware would strengthen the paper's practical relevance.

We fully agree with the reviewer that a discussion on the scalability of the demonstrated phase control strategies to high-power, fibre-based CBC systems is essential for highlighting the practical relevance of our work. In response, we have added a discussion to Section 5 (Page 9, starting at Line 311). To avoid replication, we kindly refer the reviewer to our response to Reviewer 2, Comment 1, where the revised manuscript text is provided.

- Although the authors mention "power-in-bucket" in the supplementary information, integrating more quantitative beam quality metrics (e.g., M^2 , Strehl ratio) in the main text would aid in evaluating the fidelity of the shaped beams.

We thank the reviewer for highlighting the need to more clearly present quantitative beam characteristics in the main text. In response, we have revised Section 2 (Page 2, starting at Line 63) to explicitly include the measured M^2 for the 61-beam configuration:

"A Spatial Light Modulator (SLM) was employed to modulate a continuous-wave laser from an intensity-stabilised Helium-Neon source, simulating the collimated outputs of a hexagonally close-packed array of 61 fibres (\dots , $M^2 \approx 1.626$ for the coherently combined focus, approximately 76.3% fill factor \dots)."

The M^2 is determined by fitting measured beam radius along the propagation axis to the Gaussian-beam propagation formula. We have also included the measured M^2 for the 19-beam and 127-beam configurations in the Supplementary Information (starting Page 1 Line 20):

“Fig. S1 visualises this effect, presenting Power-In-Bucket (PIB) measurements and intensity distributions along the axial direction for 19 beams ($M^2 \approx 1.634$, 84 SLM pixel radii), 61 beams ($M^2 \approx 1.626$, 42 SLM pixel radii), and 127 beams ($M^2 \approx 1.482$, 28 SLM pixel radii), all with the same fill factor.”

We have also attempted to estimate the Strehl Ratio (SR) using:

$$SR = \frac{\text{Max}(I_{\Sigma})}{\sum_n \text{Max}(I_n)}$$

where $\text{Max}(I_{\Sigma})$ denotes the peak intensity of the coherently combined beam focus, and $\text{Max}(I_n)$ denotes the peak intensity at focus of the n -th individual channel. However, due to the limited dynamic range and contrast of our camera, we were only able to (reliably) measure the SR for the 19-beam configuration, which yields a normalised value of approximately $75.0 \pm 0.6\%$. For higher channel-count configurations, including the 61-beam array used throughout the manuscript, the individual $\text{Max}(I_n)$ values could not be accurately measured, as the peak signal levels approached the noise floor, making the data unreliable. Our group forms the theory-focused part of a broader collaborative effort aiming to devise a high-power, high-brightness CBC system with extended capabilities. As such, our primary emphasis has been on PIB measurements. However, following the reviewer’s valuable comment, we recognise the importance of including other beam characteristics such as SR. We are currently reconsidering the layout of our experimental setup to enable accurate SR measurements (likely by including a pinhole-photodiode detector) for future work. However, we regret that a reliable SR measurement cannot be included in the manuscript due to current hardware limitations.

- A few minor grammatical errors and inconsistent figure referencing (e.g., “Figures 5(c4), Figures 5(d4), Figures 5(c3)”) should be polished for clarity.

We would like to express our sincere gratitude to the reviewer for bringing our attention to the typographical error and figure referencing issues presented in the manuscript. We have carefully re-examined the manuscript multiple times and made the necessary corrections to address all identified grammatical errors and figure referencing issues.

Addressing the points above will enhance its clarity and practical impact.

We would like to express our sincere gratitude to the reviewer once again for the insightful and constructive feedback. We hope that the revisions made in response will meet the high standards required for publication.